# A blueprint for biomolecular condensation driven by bacterial microcompartment encapsulation peptides

Daniel S. Trettel [1], Cesar A. López[2], Eliana Rodriguez[1], Babetta L. Marrone [1] & Cesar Raul Gonzalez-Esquer[1] ✉

Bacterial microcompartments are protein organelles with diverse metabolic capabilities. Their functional diversity is determined by an enzymatic core that is sequestered within a structurally conserved protein shell architecture. Segregation of protein cargo into the bacterial microcompartment is enabled by encapsulation peptides, which are short helical domains fused to core proteins through a disordered linker. Here, we investigate how encapsulation peptides drive multicomponent cargo assembly into biomolecular condensates. In vitro experiments supported by molecular dynamics simulations demonstrate the importance of both conserved hydrophobic packing and electrostatic interactions in stabilizing trimeric encapsulation peptide bundles. Topological rearrangements of encapsulation peptide domains can drive programmable liquid- or gel-like partitioning in vitro and in vivo. This partitioning is found to be encapsulation peptide-specific, modular, and can co-assemble at least three fluorescent reporters. In summary, we describe the molecular features necessary to drive biomolecular condensation using a widespread peptide tag. This work can serve as a blueprint for implementing encapsulation peptide biotechnology across diverse applications.

Bacteria are increasingly appreciated to dynamically organize their internal components[1,2]. One important strategy for bacterial organization is the segregation of their metabolism into organelle-like architectures, which can be found as lipid or protein membrane-bounded or unbounded (i.e., condensates)[3]. A widely studied prokaryotic organelle, called bacterial microcompartment (BMC), consists of protein polyhedra made of an outer protein shell which encases enzymatic cargo within[4]. These organelles are widespread across taxa[5] and can serve catabolic (called metabolosomes) or anabolic (called carboxysome) roles. BMCs have been proposed as a promising resource of structural (scaffolds, nanoreactors) and functional (substrate channeling, toxic chemical sequestration) motifs for their use in biotechnology[6–10]. Recent work has connected the biogenesis of some bacterial organelles to phase-separated intermediates[11–14]. This link ties protein condensation phenomena

(specifically through liquid-liquid phase separation mechanisms [LLPS])[14] to numerous critical cell functions like carbon fixation, cell division, and organelle positioning[1,15–23].

The early stages of BMC biogenesis have been suggested to make use of LLPS mechanisms, for example, in carboxysomes[24]. In β-carboxysomes, the scaffold CcmM first nucleates liquid-like droplet formation[12,17,25] of itself, Rubisco, and CcmN (i.e., procarboxysome)[26] prior to shell recruitment by an encapsulation peptide (EP) in the C-terminus of CcmN[27,28]. EPs are terminal amphipathic helices, typically 18 residues in length, fused to cargo by a poorly conserved disordered linker[29–31]. The hydrophobic residues along one face of EPs have been shown to be a common motif critical for successful microcompartment packaging[32]. This arrangement, as a single binding domain, does not strictly adhere to a canonical stickers and spacers (interaction motifs and linkers) framework commonly observed in biomolecular

[1]Los Alamos National Laboratory, Bioscience Division, Microbial and Biome Sciences Group, Los Alamos, NM, USA. [2]Los Alamos National Laboratory, Theoretical Biology and Biophysics Group, Los Alamos, NM, USA. ✉e-mail: crge@lanl.gov

condensation[33]. Metabolosomes have been similarly tied to using LLPS to guide assembly in vivo[34], presumably through their EPs, resulting in the aggregation of protein cargo to each other and to the interior of the microcompartment shell. Notably, α-carboxysomes have been found to likewise utilize liquid intermediates to facilitate assembly by local condensation of the disordered scaffolds CsoS2, Rubisco, and shell components to trigger an assembly cascade without an identifiable EP[13,18]. EPs have been observed to form in vivo aggregates[35,36] but have not been explicitly studied for their ability to trigger biomolecular condensation of fused cargo[31] for biotechnological applications.

In this work, we study the self-assembly and condensation propensity of EPs. First, we demonstrate that the model EP from the aldehyde dehydrogenase (PduP) from *Salmonella enterica* fused to an mNeonGreen cargo reporter can trigger phase separation of fused cargo in vitro. Additional protein components can be assembled into condensed droplets if they contain an EP. In vitro experiments supported by molecular dynamics (MD) simulations demonstrate the importance of both conserved hydrophobic packing and electrostatic interactions in stabilizing trimeric EP bundles. Topological rearrangements of EP domains can exert control over liquid- or gel-like partitioning in vitro and in vivo. This partitioning is found to be EP-specific, modular, and can co-assemble at least three fluorescent reporters into liquid-like foci in live bacteria. In sum, we provide a blueprint of the molecular features necessary to drive biomolecular condensation using the widespread EP peptide tag. This work paints a drastically different view of EP domains and demonstrates a condensation system for the assembly of disparate metabolic machinery in bacteria.

## Results

### The PduP EP can drive cargo accumulation into biomolecular condensates

Our investigation started with the model PduP EP from the *Salmonella enterica* propanediol utilization (Pdu) BMC. This EP has been established to be important in cargo packaging within BMC shells[30] and known to exhibit an α-helical structure by NMR and circular dichroism[31,37]. We fused this 18-residue EP, along with its native 18-residue linker (Fig. 1A), to the N-terminus of mNeonGreen[38] (EP-mNG) to act as a reporter system and purified it alongside wild-type mNG. Biomolecular condensation is commonly triggered by adding molecular crowding agents to samples[39]. As such, we titrated polyethylene glycol (average MN 2000; PEG2k) into solutions of mNG and EP-mNG at 20 μM and measured sample turbidity as a proxy for aggregation. Turbidity is commonly used to assess biomolecular condensation as particles, like condensates, refract light, and the resulting signal serves as an indirect measure of their number and size[40]. We observed no changes in turbidity for mNG while EP-mNG turbidity increased in a sigmoidal fashion with an inflection at 17.2% ± 0.6% (w/v) PEG2k (Fig. 1B). We then chose 20% (w/v) PEG2k as a constant and assayed protein concentration for turbidity changes (Fig. 1C). Again, mNG turbidity remained constant while EP-mNG increased with an approximate inflection at 33.1 μM ± 0.4 μM.

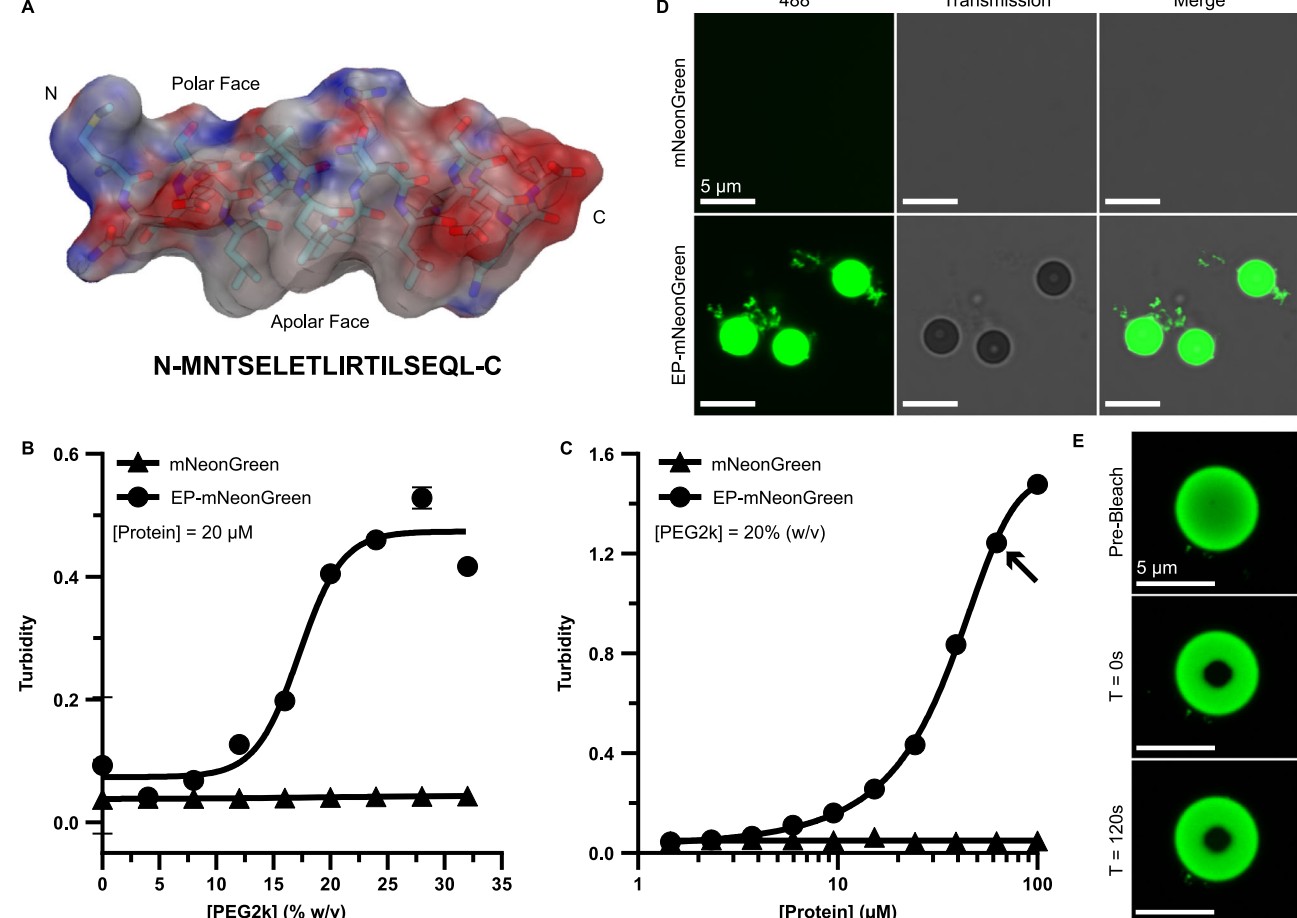

**Fig. 1 | The encapsulation peptide domain from PduP triggers biomolecular condensation of cargo. A** The model EP from PduP is an amphipathic, 18-residue α-helix fused to the amino terminus of the protein cargo by a disordered linker. The residues depicted in the helix representation are shown below it. Turbidity assays are used to assess particle formation from EP fusions as a function of **B** molecular crowder PEG2k and **C** protein concentration. Error bars represent the standard deviation from the mean from 4 replicates. The arrow in (**C**) indicate the samples taken for imaging. **D** Confocal imaging reveals the presence of condensates in solution for samples containing EP fused cargo. **E** Photobleaching experiments show minimal/no fluorescence recovery.

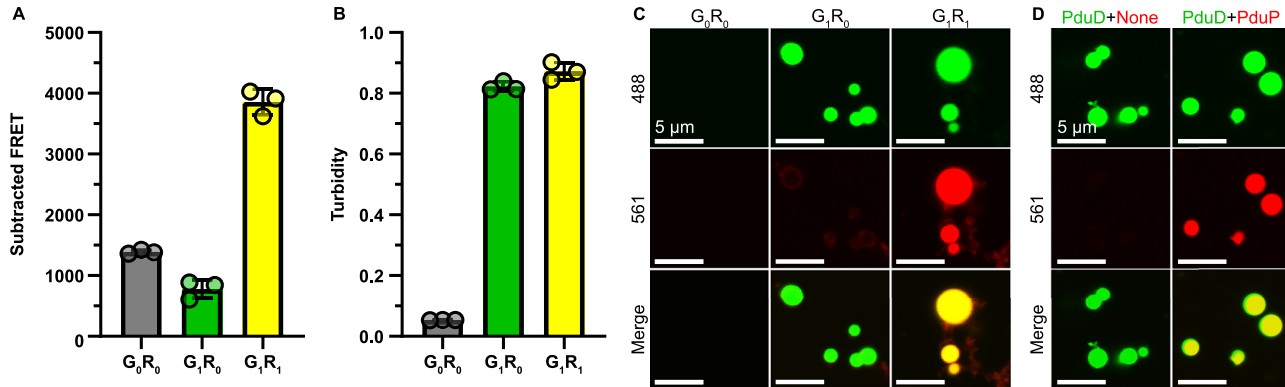

**Fig. 2 | Specific in vitro co-assembly of multiple cargoes with encapsulation peptides. A** Fluorescence resonance energy transfer (FRET) was used to assess assembly of a donor mNeonGreen (G) and acceptor mScarlet-I3 (R) with (1) or without (0) N-terminal EP fusions. FRET was calculated by subtracting the emission of mNeonGreen by itself at the peak acceptor emission wavelength (590 nm). Significant FRET only occurred when both cargoes have encapsulation peptides. Error bars represent the standard deviation from the mean from 3 replicates. **B** Turbidity assays of samples confirm particle formation only when at least one

cargo had an EP. Error bars represent the standard deviation from the mean from 3 replicates. **C** Confocal imaging showed that specific co-assembly of mNeonGreen and mScarlet-I3 only occurred when they both had EP fused to their N-termini. **D** Similarly, co-assembly is observed with heterotypic mixes of EPs, with mNeonGreen fused to the EP from PduD (green text) condensing with mScarlet-I3 when it is fused to the EP of PduP (red text). In both (**C**) and (**D**), mNeonGreen and mScarlet-I3 were excited with the 488 and 561 nm laser, respectively.

**Fig. 3 | Encapsulation peptides rely both on hydrophobic packing and a newly identified salt bridge network. A** A distance matrix of two assembled PduP EPs shows that the closest interactions are from hydrophobic residues (black outlines) along the apolar face. **B** Computational models of EP dimers suggest that they prefer an antiparallel orientation, which results in a salt bridge network between E7 and R11. The heteroatom distances are represented by dashed lines with distances

(in Angstroms) calculated in PyMol. Several mutants were made, and their condensation propensities were screened by measuring turbidity as a function of **C** NaCl for electrostatic screening and **D** total protein concentration. These data demonstrate that EP self-assembly is sensitive to ions and condensation can be negatively impacted by disrupting the salt bridge and hydrophobic core. Error bars represent the standard deviation from the mean from 4 replicates.

The above data suggested that the EP tag can result in particle formation in solution. However, turbidity assays cannot differentiate between cargo aggregation and condensation[41]. We investigated this by imaging turbid samples (Fig. 1C, arrow) with laser scanning confocal microscopy. Imaging showed no structures formed by mNG itself (Fig. 1D). In contrast, EP-mNG was able to spontaneously demix and form spherical droplets in solution (alongside some amorphous material), which is suggestive of phase separation coupled to percolation (PSCP)[42]. Biomolecular condensates that form through PSCP should display changes in their viscoelastic material properties as a function of time[42]. We tested the material state of these droplets by performing a fluorescent recovery after photobleaching (FRAP) experiment to monitor the recovery of fluorescence[43]. If liquid-like, droplets will experience quick recovery of fluorescence (typically <60 s timescale) in bleached zones owing to rapid exchange and equilibration of bleached and unbleached molecules. The droplets formed by EP-mNG, however, did not experience significant recovery over a qualitative 2-min sampling (Fig. 1E), indicating slow internal dynamic rearrangements indicative of a gel-like state. The droplets were further commonly observed in arrested fusion states that never resolved (Supplemental Fig. 1A) and likewise reminiscent of gelation. We also found that if our usual order of component addition was inverted (protein from a concentration stock added last) then these mid-fusion assemblies or larger aggregates, which appear as numerous unresolved droplets, were more commonly observed (Supplemental Fig. 1B). These data suggest that these assemblies undergo rapid networking which cannot sufficiently resolve into distinct droplets when triggered from a concentrated protein stock. This, and their inability to recover quickly from bleaching, suggests that EPs enable PSCP to form physical microgels and that transitions in their viscoelastic properties happen on relatively short timescales.

## The EP domain can drive specific co-assembly of multiple species

We next sought to understand if EP-driven condensates can include more than one component. We tested this by mixing purified mNeonGreen (G) and mScarlet-I3[44] (R) either with ($G_1$ or $R_1$) or without ($G_0$ or $R_0$) an EP fusion and measured the results by fluorescence resonance energy transfer (FRET), turbidity, and imaging methods (Fig. 2A–C, respectively). Mixed samples with no EPs ($G_0R_0$) were unable to produce a significant FRET or turbidity signal and confocal imaging found no condensates in solution. Samples which included an EP only on mNeonGreen ($G_1R_0$) showed a slight decrease in FRET signal and high turbidity. Imaging results indicated that the turbidity was a result of mNeonGreen condensates with a weak outer halo of mScarlet-I3. The slightly lower FRET signal was likely due to mNeonGreen condensation in solution, which limited mScarlet-I3 interactions. Lastly, samples with EPs on both proteins ($G_1R_1$) produced the highest FRET signal and turbidity equal to $G_1R_0$ samples. Confocal imaging showed that $G_1R_1$ condensates were formed from co-assembled mNeonGreen and mScarlet-I3, which agrees with FRET data. EP-dependent co-assembly was also tested with mNeonGreen and mScarlet-I3 cargoes fused to different EPs (Fig. 2D). Here, mNeonGreen fused to the EP of PduD (part of the signature enzyme complex PduCDE in the Pdu BMC)[45] and mScarlet-I3 fused to the EP of PduP were mixed together to test their co-assembly. Microscopy shows that the PduD EP can condense mNeonGreen alone and support co-assembly with the PduP EP. Together, these data demonstrate that different EPs can drive specific homo- and heterotypic co-assembly of multiple cargoes in vitro.

## EPs assemble through hydrophobic packing and a specific salt bridge

Our model system gives us an opportunity to screen EP self-assembly through simple turbidity measurements. EPs have traditionally been thought to act solely through relatively non-specific hydrophobic

packing interactions predominantly along their apolar face (Fig. 1A)[32,46]. We confirmed this by performing multiscale (integrated coarse-grained and fully atomic representation) MD simulations by inspecting the dimerization process of two EP helices. Using the Martini 2.2 force field, self-assembled structures were backmapped into their correspondent fully atomic representation and equilibrated at microsecond range using the charmm36m force field[47,48]. A residue contact map of tightly associated helices agrees with the notion that hydrophobic residues, for instance I10, form the closest and most abundant contacts between bound helices (Fig. 3A, black boxes) and have been previously implicated with being critical for cargo encapsulation within microcompartment shells but not self-assembly[32]. A closer look at simulated EPs also identified an antiparallel salt bridge network formed by E7 of one helix and R11 from the other spaced by exactly one helical repeat (Fig. 3B). We tested the importance of these residues by making several point mutations and compared their condensation propensity as a function of NaCl concentration (to disturb the salt bridge) and protein concentration. The mutations we chose to assay are R11K (conserved mutation), R11A (removal of salt bridge), E7D (side chain one carbon shorter), I10S (disrupt hydrophobic core), and a double E7R/R11E mutant which swaps the relative orientation of the salt bridge.

A salt titration assay confirmed that all EP mutants, to varying extents, resulted in turbidity and that turbidity is negatively impacted by increasing salt (Fig. 3C). At low salt saturation, the R11K and E7R/R11E mutants performed similarly to the wild-type PduP EP, confirming that functionality is conserved. However, as expected by its hydrogen bond strength, high salt concentration had slightly more effect on the R11K mutant. The E7D mutant, which may simply weaken the salt bridge, was less robust than the R11K mutant but more robust than the R11A mutant, which completely removed that interaction. Further, the inflection points of all salt bridge mutants were shifted left towards lower concentration (140 mM for WT versus 80-100 mM for mutants), revealing a heightened salt sensitivity. Among these, the double E7R/R11E mutant demonstrated the greatest dependence on NaCl concentration, suggesting that the relative orientation influences assembly and other factors may be important. Confocal imaging of the wild-type PduP design shows that increasing salt results in reorganization of droplet-like condensates into aggregates, which then dissolve at a high enough concentration (Supplemental Fig. 2).

These trends were further reflected when titrating protein concentrations (Fig. 3D). Again, WT and R11K performed similarly, while salt bridge mutants showed an intermediate self-assembly propensity. The I10S mutant had no assembly propensity. The double E7R/R11E mutant performed similarly to WT, indicating that, under these assay conditions, the salt bridge is sufficiently recapitulated in a reverse orientation and promotes condensation. These results indicate that the computationally predicted antiparallel salt bridge is real and impacts self-assembly strength of EP domains. The most critical residue we tested, however, is I10 within the hydrophobic core.

## Metabolosome EPs are functionally distinct from carboxysome EPs

Most BMCs use EPs to accumulate cargo within their shells[29]. This includes β-, but not α-, carboxysomes, which use the C-terminus of the protein CcmN to drive associations between CcmM-RbcLS coacervates and the shell[28]. The C-terminal extension of CcmN proteins have been regarded as EPs because they literally link cargo-shell interactions. This is a very different arrangement for cargo inclusion interactions compared to metabolosomes, wherein EPs typically exist fused to enzymatic cargo as opposed to CcmN, which acts as a discrete intermediary between enzymatic cargo and the shell. Sequence comparisons of several candidate metabolosome and CcmN EPs showed that metabolosome EPs largely conserve the salt bridge related residues we have identified with a sequence spacing corresponding to one helical repeat

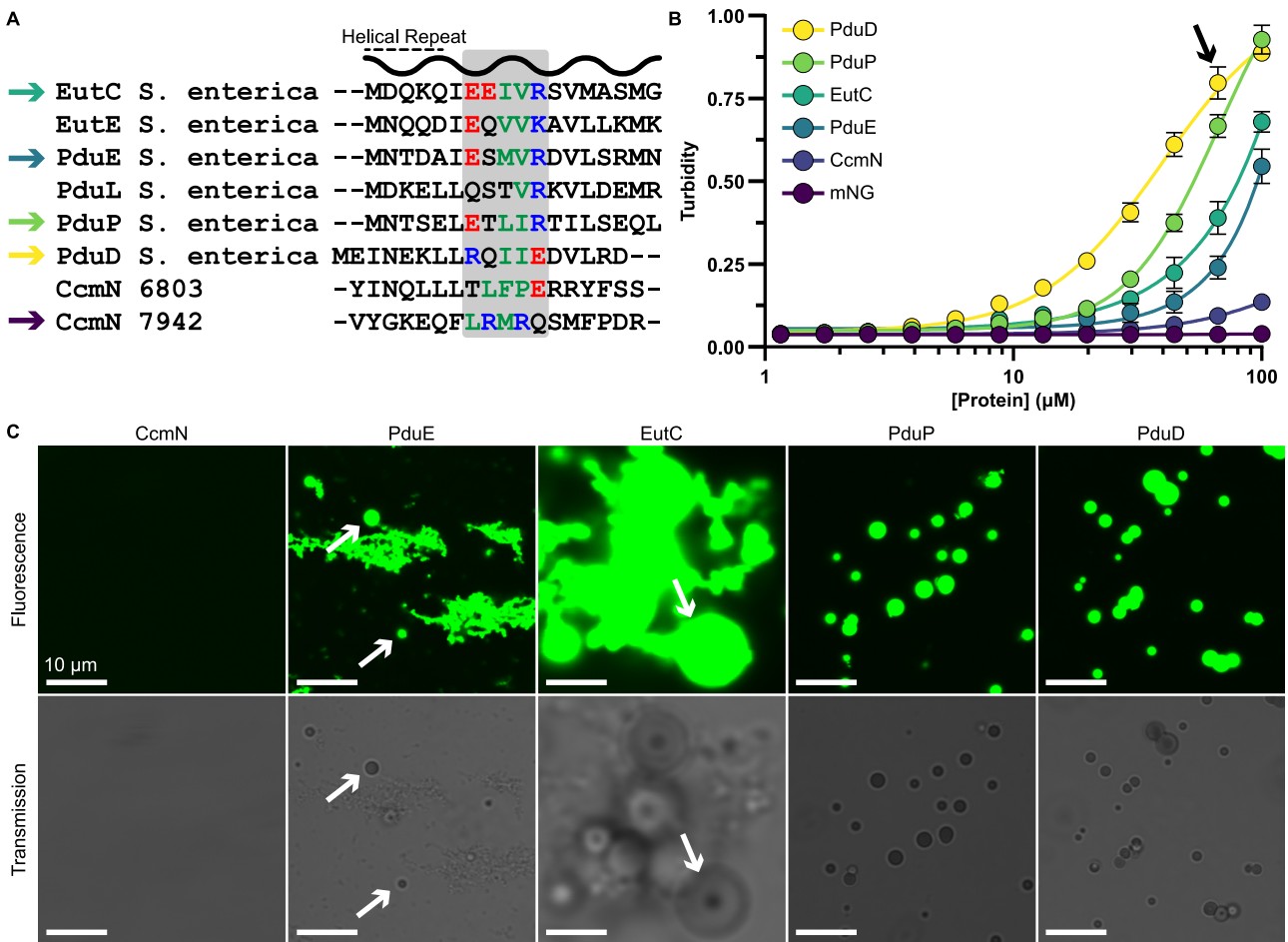

**Fig. 4 | Metabolosome EPs have distinct sequence elements which carry functional consequences. A** An alignment of several EPs revealed that metabolosome EPs carry oppositely charged residues separated by one helical repeat. These elements are not maintained in carboxysomal CcmN sequences. Basic residues are colored blue, acidic are red, and apolar are green. **B** A turbidity assay demonstrated that these sequence differences result in CcmN being unable to condense cargo under conditions that trigger condensation for metabolosome EPs. Error bars represent the standard deviation from the mean from 4 replicates. **C** Aliquots from turbid samples (arrow in **B**) were imaged with confocal microscopy. Spherical droplets were observed in all metabolosome samples, to varying extents (arrows) and most clearly defined in PduP and PduD samples. CcmN did not drive droplet structures.

(Fig. 4A). The EPs of CcmN, however, do not carry these sequence elements. We investigated this further by screening the ability of different EPs fused to mNeonGreen to trigger biomolecular condensation of this cargo. In addition to our PduP model, we also selected PduD (swapped charges, similar to our E7R/R11E PduP mutant), PduE (reportedly not sufficient for cargo encapsulation[45]), EutC (different BMC), and CcmN (from *Synechococcus elongatus* PCC 7942)[26,28]. All metabolosome EPs (PduDEP, EutC) were found to result in cargo condensation to varying degrees (Fig. 4B). PduD was the most robust, followed by PduP, EutC, and finally PduE. Meanwhile, the CcmN EP was not able to trigger significant cargo aggregation. Protein droplets were likewise found to exist in all metabolosome EP samples, albeit in co-existence with fibril-like clusters in the case of PduE and EuC, but not with CcmN (Fig. 4C). These self-assembly trends were likewise recapitulated with in vivo overexpression, with CcmN unable to drive cargo partitioning (Supplemental Fig. 2A, B).

## EPs associate as trimeric complexes

Our computational calculations demonstrated that EPs can form a specific salt bridge when paired in an antiparallel orientation (Fig. 3B). This association, however, splays open the hydrophobic core between the helices, indicating additional association capacity. We experimentally tested this by fusing serial repeats of the PduP EP and its native linker (EP1, EP2, EP3) to the N-terminus of an mNeonGreen cargo (Supplemental Fig. 3A). More repeats were not investigated due to issues cloning higher copy numbers, likely owing to the repetitious nature, despite our best efforts to mitigate sequence repeats. We hypothesized that this synthetic arrangement of intrinsic valency, mimicking a more classic stickers and spacers arrangement, will either (1) increase binding capacity linearly or (2) form intramolecular arrangements (due to proximity and connectivity) that self-quench binding capacity and lead to diminishing returns. We assayed these designs for self-assembly propensity and found that the EP2 design moderately outperformed the EP1 while the EP3 performed similarly or slightly worse than the EP2 (Fig. 5A). Specifically, curve inflections for the EP1, EP2, and EP3 designs were found to be (in $\mu$M) $33.1 \pm 0.4$, $15.5 \pm 1.0$, and $19.7 \pm 1.9$, respectively. The morphology of all samples was also similar, and all shared the ability to form droplets across a wide range of concentrations (Fig. 5B). We note that these images alone could not establish a threshold concentration for condensation, which may presumably lay in the low micromolar to high nanomolar range depending on assay conditions. Here, increasing protein concentration increased both the size and quantity of droplets from all designs. These designs were similarly assayed for response to PEG2k concentration with identical trends observed (Supplemental Fig. 4A). These results suggest intramolecular valencies above two may quench

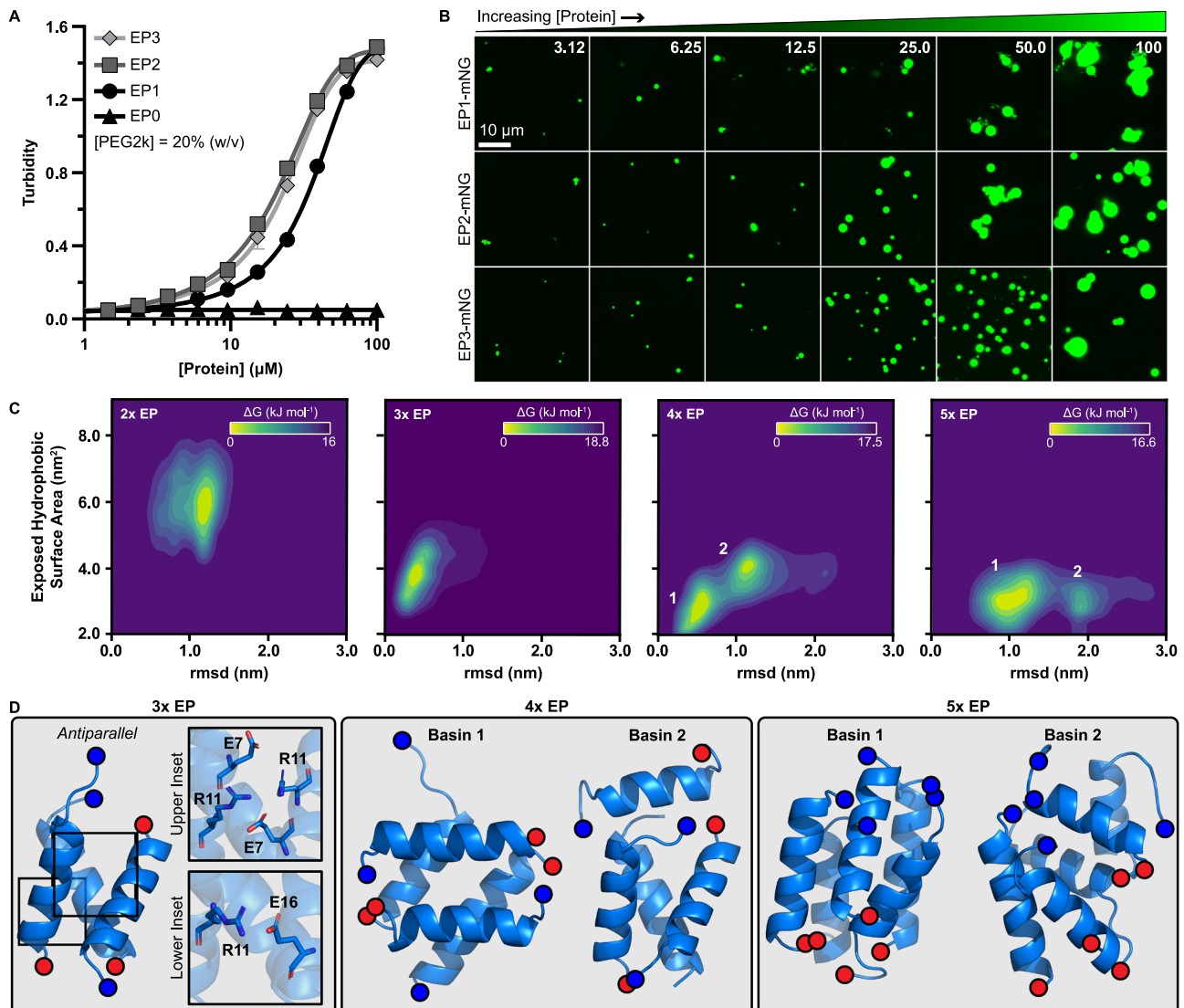

**Fig. 5 | Encapsulation peptides self-assemble in a multivalent manner.**
**A** Turbidity was measured as a function of protein concentration for synthetic EP designs where EPs were fused in serial to the cargo N-terminus. Additive copies had diminishing results. Error bars represent the standard deviation from the mean from 4 replicates. **B** Despite differences in sequence topology and condensation propensity, confocal imaging showed all designs formed similar condensates in solution across a range of concentrations (in μM). **C** MD simulations were conducted with two, three, four, or five EP helices to study their preferred assembly state as a function of exposed hydrophobic surface area and root mean square deviation (rmsd). Simulations with two or three EPs coalesced to one basin with low rmsd, while those with four or five showed two dominant populations. **D** Structural analysis of these basins revealed that EP assembly is associated with salt bridge interactions along their solvent exposed surfaces and maximize hydrophobic packing.

additional associations (Supplemental Fig. 4B–D), thereby providing evidence for associations greater than a dimer.

To understand the higher order association mechanism, we resorted to MD calculations to analyze the saturation binding capacity of EPs. In addition to dimeric association, we also tested simulations of 3, 4, or 5 associated PduP helices with the enhanced coarse-grained (CG) approach to access information on systems we could not experimentally create. The equilibrated CG configurations were transformed to atomic resolution and re-equilibrated at microsecond range, data which is summarized as a function of both positional variance (RMSD) and exposed hydrophobic surface area (Fig. 5C). Simulations with 3 helices coalesced towards one single basin composed of two overlapping species in parallel and antiparallel configurations, with the orientation of one helix being swapped between the two. These configurations closely align structurally (Supplemental Fig. 5A), agreeing with the low RMSD and exposed hydrophobic surface area (Supplemental Fig. 5B), but show slightly different salt bridge networks

between E7, R11, and E16 (Supplemental Fig. 5C). Meanwhile, configurations composed of 4 or 5 helices formed two basins. Representative configurations for the different basins are provided in Fig. 5D, highlighting the helical association. In particular, the 3-helix configuration remained very stable and formed a tightly packed bundle, a result which was observed in four different replicates. In contrast, other stoichiometries (4 and 5 helices) led to unstable bundles that transition between configurations. In either case a trimeric bundle was observed in basin 2 which led to a single or dimeric helix that is unstably associated (Fig. 5D). Dimerization also leads to a configuration that was stabilized by a series of hydrogen bonds (as described before); however, the abundance of exposed hydrophobic core makes this configuration rather unstable and hypothesized as a metastable intermediate structure towards trimerization (Supplemental Fig. 6).

We further investigated whether trimerization would lead towards higher ordered configurations by self-assembling stable trimeric helices at different concentrations in silico. Aided by the latest

Martini 3 force field[49], we characterized the self-assembling propensity of EPs in the trimeric configuration (summarized in Supplemental Fig. 7). Even using high peptide concentrations (20 μM), our data indicates that trimeric bundles (with no cargo attachment) barely associate in solution. In fact, total peptide concentration must be at least one order of magnitude higher to observe larger assemblies. These insights led us to conclude that optimal trimeric bundles are expected to initially guide the association of EP domains, while localization would further enhance the interaction of these bundles, cooperatively affecting the condensation process.

The lack of a conclusive condensation enhancement by the EP3 construct in our experiments (Fig. 5A) led us to hypothesize that self-quenching may reduce the interaction of the tandem repeat. CG simulations were performed comparing EP2 and EP3 designs which show that covalently connected EP sequences can self-interact, effectively reducing the number of available binding sites for network extension (Supplemental Fig. 8). The computed equilibrium between different configurations indicates that the EP3 design can self-bind to the extent of completely closing the number of available binding sites, partially explaining the condensation results of our tandem sequence designs. The EP2 construct would always have additional networking capacity even if the domains intramolecularly dimerize (Supplemental Figs. 4B–D and 8). The EP3 design, in contrast, may have one more interaction site available, however, the self-blocking can potentially hinder further network extension.

## EP topology can alter partitioning and material state in vivo

Our in vitro model system thus far has included strictly monomeric proteins, which differs from those natively tagged with EPs and may not reflect physiological reality. For instance, PduP is a homotetramer[50] and PduCDE, which has EPs on PduD and PduE, is a dimer of trimers[51]. We investigated the effects of binding site valency by again implementing our serial EP designs fused to a monomeric mNeonGreen against a tetrameric E2-Crimson[52] (hereafter Crimson) to differentiate between intrinsic and emergent valency, respectively (Fig. 6A). We sought to explore the functional differences between these two arrangements in vivo to gain a proper understanding of EP-driven assembly in living cells. Accordingly, we overexpressed these designs in *Escherichia coli* and used laser scanning confocal microscopy to image their phenotypes in vivo (Fig. 6B). Both the wild-type mNeonGreen and Crimson showed a uniform fluorescence inside of cells. Meanwhile, increasing the EP copy number on mNeonGreen similarly increased the segregation of multiple (typically 2+) mNeonGreen cargo foci within the cells. This effect was also observed with a single EP copy fused to Crimson cargo, which formed single elongated foci. We quantified this effect by calculating a partition ratio (the intensity of background subtracted foci divided by non-foci areas of the same cell) which is analogous to how much denser foci are compared to the dilute areas of the cell (Fig. 6C). Quantification in this way gives the mNeonGreen and Crimson controls a median partition ratio close to 1.0, indicating uniform fluorescence distribution throughout the cell. This analysis further shows that, in contrast to our in vitro tests, additive copies of EP domains increased partitioning up to ninefold over the mNeonGreen control for the EP3 design. This effect was exacerbated in the EP-Crimson design, which was able to achieve a median partition ratio of over 500, making it nearly all-or-nothing. Overall, these results demonstrate the importance of valency in driving cargo networking and that our serial repeat design likely prefers intermolecular, and not intramolecular, associations in vivo. We believe these differences compared to our in vitro experiments may stem from the dense cytoplasmic milieu inhibiting specific intramolecular associations and promoting intermolecular networking.

We next assayed the material state of EP-driven foci within cells by measuring in vivo FRAP[43]. Here, single foci within cells were bleached with a laser pulse and their fluorescence recovery was monitored with 0.5 s intervals. FRAP experiments were focused on comparing the relative differences between EP2 and EP3 designs (Fig. 6D) due to quick recovery of EP0 samples and EP1 foci being hard to find. FRAP was calculated by normalizing the background (yellow circle, Fig. 6E) subtracted fluorescence of bleaching foci (red arrow) to an unbleached focus (white arrow) to see when separate foci equilibrated. The EP2 and EP3 designs were able to achieve full re-equilibration relative to unbleached foci within 2 min and with half-times of 16.2 s ± 8.5 s and 27.2 s ± 14.6 s, respectively. We note that this is slower than recovery of free fluorescent protein, which is on the order of a few seconds[53]. This contrasted drastically with results from cells expressing EP-Crimson, which demonstrated a linear and incomplete recovery profile within this timeframe, indicative of a gel-like state (Supplemental Fig. 4)[22,23,43]. The relatively large errors may stem from different distances between bleached and unbleached foci within cells, therefore leading to longer diffusion distances. Despite this, the recovery curves of EP2- and EP3-mNG were significantly different ($p < 0.001$). This experiment alone cannot determine if the difference between the EP2 and EP3 designs stem from diffusion rate differences (EP3 is larger and may migrate slower) or due to slight changes in their condensed material states. However, it would make sense that the EP3 does equilibrate slower than EP2. The EP3 design likely forms more intermolecular contacts to shift equilibria, a function of on and off rates, as informed by its increased partition ratio and thereby slowing equilibration. These results demonstrated that EP-driven condensation in bacteria was liquid-like in contrast to our in vitro results; and in equilibrium with the excluded cytosol, and not inclusion bodies, which have been demonstrated to not recover during similar experimentation[43]. Further, topological rearrangements of EPs can be used to encode for different cytosolic behavior.

## EP partitioning is specific and can partition multiple components in vivo

Overexpression of a single protein fused to the PduP EP can partition it between a condensed and dilute phase in vivo. However, more complex arrangements exist naturally, and multicomponent systems would be desired synthetically. We first tested the ability of a single mNeonGreen cargo to partition when co-expressed equally (assumed from ribosomal binding sites) with mScarlet-I3 and mTagBFP2[54]. Designs were made with mNeonGreen encoding for up to three EP repeats and given a nomenclature based on the gene ordering and number of EP repeats (Fig. 7A, top). For instance, a design with two repeats only on mNeonGreen is named $G_2R_0B_0$ (for green with two EPs, red with zero, blue with zero). Constructs were overexpressed and mNeonGreen partitioning was calculated as before. The general trend of more EP repeats leading to greater partitioning was again observed (Fig. 7B, C). However, the effect was muted compared to the single-expression designs, which we suspect stemmed from a competition for resources between the three proteins during expression. That is, the competition may have lowered the effective expression, and therefore concentration, of EP domains and reduced mNeonGreen fluorescence was observed again, confirming the specificity. In addition, the $G_3R_0B_0$ expressing cells also reveal dark spots in the red and blue scans where the mNeonGreen appeared to localize, suggesting exclusion of non-EP tagged proteins.

Next, we added in a series of equal repeats of EP tags to all three fluorescent proteins to encourage multicomponent assembly (Fig. 7A, bottom). Confocal imaging of designs again confirmed that $G_0R_0B_0$ showed no signs of visual foci (condensation or aggregation) (Fig. 7D). Calculating the partition ratio likewise gave values of near 1.0 (Fig. 7E). We also calculated a summed partition ratio (Fig. 7F) which is the sum of all background subtracted foci signals divided by the sum of the background subtracted cellular background. For the $G_0R_0B_0$ design, this value again came to 1.0, indicating no partitioning without EPs. Noticeable partitioning began to occur with the $G_1R_1B_1$ designs

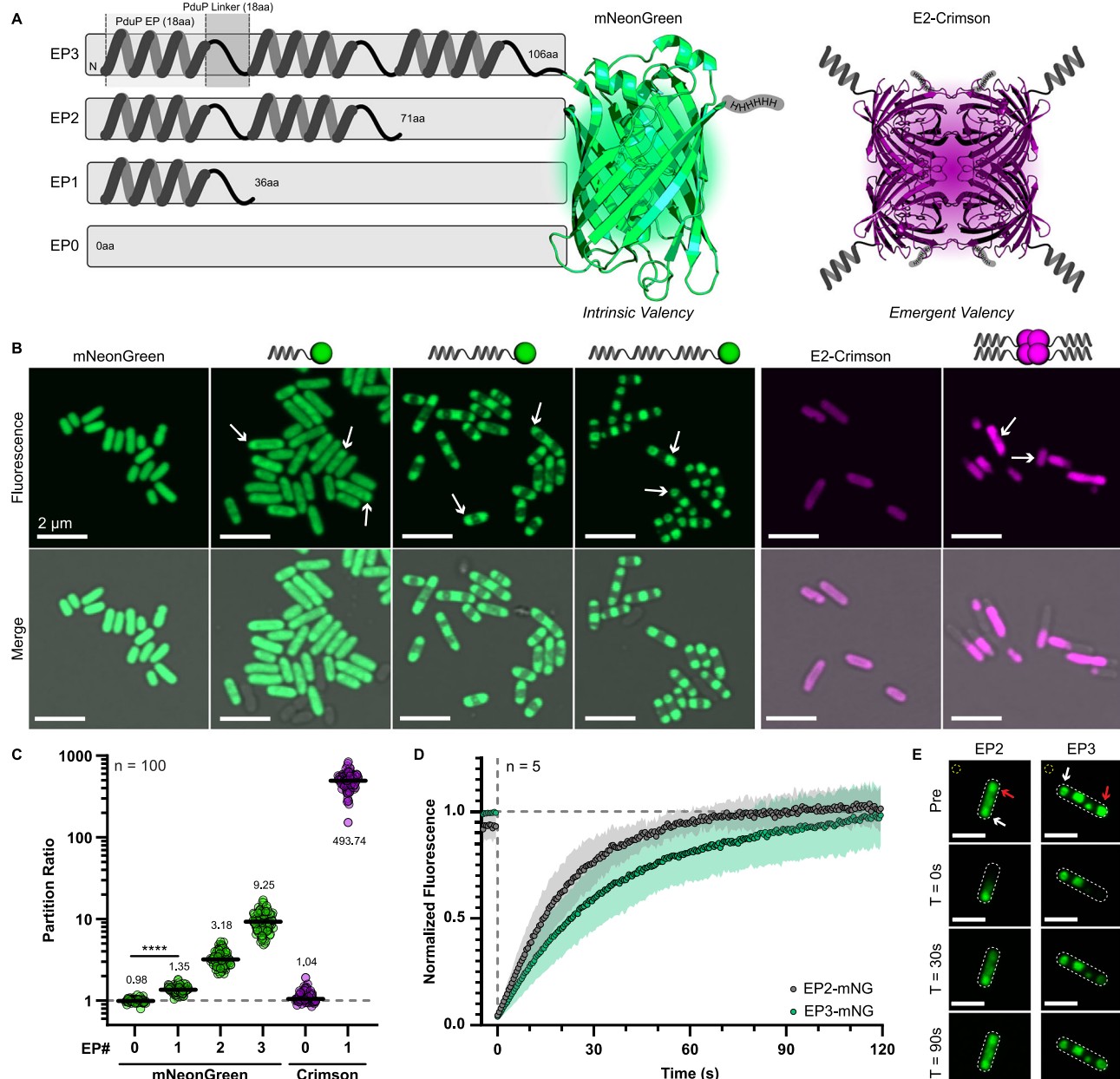

**Fig. 6 | Valency affects cargo partitioning in vivo. A** We tested the cargo partitioning performance of serial EP designs on a monomeric mNeonGreen cargo (intrinsic valency) and on a tetrameric E2-Crimson cargo (emergent valency). The mNeonGreen structure is from PDB 5LTR and E2-Crimson is represented by a tetrameric DsRed 1GGX. **B** Overexpression of designs for intrinsic valency showed that increasing valency increased partitioning ability. The emergently valent EP1-Crimson showed elongated single foci. Arrows denote several example foci within each sample. **C** Partitioning was calculated by the quotient of background subtracted foci signal over non-foci signal. This analysis confirmed that partitioning increases with additional EP domains and that emergent valency outcompetes intrinsic in all cases. Medians are denoted within each sample with the black bar. **** denotes $p < 0.0001$ given by an unpaired $t$-test. **D** Fluorescence recovery after photobleaching (FRAP) experiments showed that the EP2 and EP3 designs differed in their recovery rates, but both were liquid-like. Standard deviations are represented by the infill. **E** FRAP was calculated by normalizing the background subtracted (yellow circle) intensity of bleached foci (red arrow) to another unbleached foci (white arrow) within the same cell. Scale bar, 1 μm.

(Fig. 7D), which correlated with increased partitioning of the individual (Fig. 7E) and total (Fig. 7F) components. The partitioning becomes even more pronounced in cells expressing the $G_2R_2B_2$ designs, with cell exhibiting a highly overlapping zebra-strip patterning (Fig. 7D). The individual partitioning of components within the $G_2R_2B_2$ expressing cells was not equal (Fig. 7E) but followed the same trends as the $G_1R_1B_1$, suggesting that the characteristics of the individual proteins themselves can influence partitioning, as we found that mScarlet-I3 cargo shows a moderately increased EP-dependent partitioning ability compared to mNeonGreen (Supplemental Fig. 10). However, the total partitioning continued to increase (Fig. 7F) and suggests that the total sum, or concentration, of EP domains drives overall partitioning.

## Discussion
### Study summary
This study focused on defining the ability of EP domains to drive self-condensation of fused cargo proteins. We found that EPs resulted in cargo partitioning both in vivo and in vitro using predictive computational simulations and wet lab experiments. This self-assembly depended on hydrophobic packing and salt bridge interactions. These

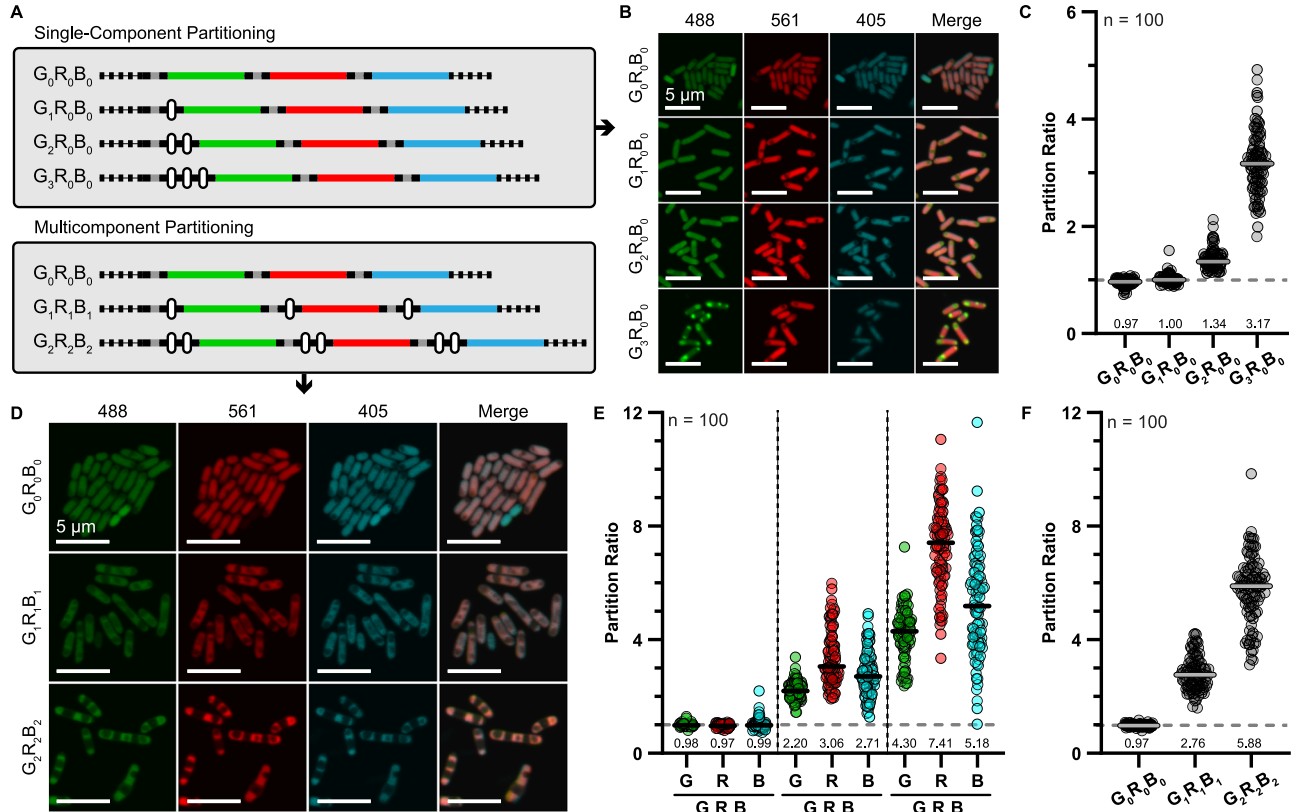

**Fig. 7 | Programmable and specific multicomponent cargo partitioning using the encapsulation peptide domain. A** Synthetic operons were designed with three different fluorescent cargos of mNeonGreen, mScarlet-I3, and mTagBFP2 to test specific single and multicomponent partitioning. These designs were given a nomenclature based on the gene order and number of EP domains fused to that cargo. Here, EP repeat elements are signified by white rounded rectangles and ribosomal binding sites by gray lines. **B** Overexpression of designs expressing EP domains only fused to mNeonGreen showed only green components in foci formed. **C** As in the single-protein designs, more EP domains resulted in increased partitioning of the EP cargo. **D** Fusing EP domains to all cargo resulted in their co-assembly into condensed "zebra-striped" bands within the cell. Calculating the partitioning of these cargoes either (**E**) separately or **F** combined again showed that additive copies enhance combinatorial cargo partitioning. For all plots, $n = 100$ cells and the median partition ratios are denoted.

interactions sum to a preference and specificity for trimeric associations. The sequence determinants are conserved among metabolosome EPs but not CcmN from carboxysomes, resulting in CcmN's inability to promote self-condensation. Further, cargo partitioning was entirely modular and solely dependent on the inclusion of the EP domain both in vitro and in vivo.

## Critical residues in EP self-assembly

Our data elucidates design principles for how EPs self-assemble. Prior work had found that the hydrophobic core is critical to encapsulation within microcompartment shells[32], and we confirm that same propensity applies for EP self-assembly too. In addition, we discern previously unidentified salt bridges that also stabilize EP self-assembly. These salt bridges can form in a multitude of arrangements - our MD simulations suggest a preference for trimeric associations for the PduP EP. Salt bridge formation can form between various residues, including E7 and R11, in both parallel and antiparallel configurations. In agreement with the MD calculations, our in vitro data showed that, among the several PduP EP mutants we screened, I10 was the most critical residue in enabling condensation. Supported by the proposed trimeric packing mechanism, mutation of the central isoleucine nearly removed all condensation ability, while disruptions to salt bridge related residues only moderately decreased condensation propensity and demonstrate that salt bridge formation alone is not sufficient to drive EP associations. As such, we propose that EP trimerization is initially enhanced by non-specific hydrophobic packing, which then

enables accessory salt bridge formation for further stability and specificity. However, EPs themselves are not sufficient for condensation alone at the concentrations we experimentally validated (Supplemental Fig. 7). Rather, we believe the trimeric bundles initially guide protein association to develop a high local concentration, which further enhances their interaction, cooperatively promoting the condensation process. These findings refines conventional notions where EPs are regarded as any short amphipathic α-helices separated from cargo by a flexible and poorly conserved linker within a microcompartment context[29,30]. Instead, we identify sequence elements that guides specific higher order networking.

## Packaging peptides by function but not by form

We tested the self-assembly of several different commonly cited EPs from PduD, PduE, PduP, EutC, and CcmN[28,30,45,55–57]. We were motived by the notion that β-carboxysome EPs (encoded on CcmN) act strictly as intermediaries between the procarboxysome and the shell, while metabolosome EPs (encoded on PduDEP, EutC) reportedly act as a single module to aggregate and encapsulate cargo. Further, CcmN is present in very low copy numbers in β-carboxysomes[58] while metabolosome EPs dominate interior interactions between themselves and the shell[59]. We noted that the sequence motifs in metabolosome EPs found to be important were missing from CcmN (Fig. 4A). We provide experimental evidence that the CcmN EP does not result in clear condensation under the same conditions as metabolosome EPs (Fig. 4B), which is congruent with its physiological role as a low-

occupancy tether[26,58]. Our imaging also demonstrates metabolosome EPs can broadly trigger biomolecular condensation of fused cargo (Fig. 4C). This data makes us believe that, while the CcmN EP may be able to guide cargo encapsulation into synthetic β-carboxysome shells[60], it cannot be used to guide robust cargo packaging alone since it lacks the ability to self-assemble. Using metabolosome EPs, in contrast, would likely be a more robust packaging strategy for building synthetic microcompartments or shell-free enzyme partitions. EPs are still an apt name for both metabolosome and β-carboxysome EPs given their ability to package cargo within BMCs. However, a sub-categorization of "assembly peptides" may be more appropriate for metabolosome EPs, which have a second role in self-assembly and could better reflect their functional diversity. In any case, an updated and more rigorous bioinformatic investigation into EP domains to better grasp their functional landscape is warranted.

Subtle differences also exist within metabolosome EPs. We observed that all our selected EPs could trigger biomolecular condensation, but the success varied (Fig. 4B). The EP from PduD is found to be the most robust, followed by PduP, EutC, and PduE. This agrees with recent observations that PduCDE, led by PduD EP, triggers hierarchical assembly of accessory cargo components in Pdu BMCs[34]. Importantly, comparing the EPs from PduD and PduP (as well as our E7R/R11E mutant) shows that the relative positioning of oppositely charged residues can be swapped as long as they maintain a sequence distance of one helical repeat. Still, other residues not explicitly tested in this work may also contribute to EP condensation. For instance, our MD calculations identified that E16 of the PduP EP may help to stabilize bundles in combinations with E7 and R11 (Supplemental Fig. 5). This notion is tentatively supported by our PduP E7R/R11E mutant, which would create at least one electrostatic clash (Fig. 5D) that may explain its heightened salt sensitivity (Fig. 3C). Further, the E16 position is not conserved among metabolosome EPs (Fig. 4A) and correlates with their condensation phenotypes. For instance, PduD and PduP have an acidic residue at that position while EutC, with weaker condensation strength, has a polar serine. The weakest EP we tested, from PduE, has an acidic residue at that position. This may also explain why EutC and PduE form more fibril like structures alongside some condensates (Fig. 4C). Additionally, the weak ability of the PduE EP to trigger self-condensation mirrors its prior described inability to package cargo within microcompartments and that these two functions may be related[45]. While our data presented cannot explicitly explain the differences between different EPs, we speculate that, in addition to specific residues like E16, it may also relate to helical stability by flanking residues. For instance, circular dichroism of synthetic produced PduD and PduP EPs has shown that the PduD is more helical in conformation[37]. A more stable helix could increase the entropic contribution during binding as we have observed in simulations on the PduP EP. As such, rationally designing stable helices, in conjunction with the critical sequence elements we identify, may provide a strategy for producing non-native EPs with tunable properties[32].

## Speculation on the physiological role and dependencies of EP-driven condensation

One surprising finding was that the organization of EP valency (intrinsic versus emergent) drastically changes cargo partitioning and can mediate their material state in vivo. Intrinsic valency refers to when a molecule has more than one binding site encoded within that molecule, while emergent valency arises from the assembly of multiple subunits, thereby increasing the number of binding sites per molecule (diagramed in Supporting Fig. 4B–E)[42,61]. Serial additions of EPs onto monomeric mNeonGreen (intrinsic valency) resulted in fluid foci with moderate partitioning abilities, while a single EP on tetrameric Crimson (emergent valency) turbocharged partitioning and significantly lowered liquidity, demonstrated by dramatically slower photobleaching recovery in vivo. Our EP-Crimson design exemplifies emergent

valency that is more reflective of all characterized EP fused enzymes. These findings suggest that BMC cargoes, which largely demonstrate emergent valency, can self-nucleate at a low saturation concentration and sequester nearly all cargo components during biogenesis. In native instances, the size of these assemblies may be determined by the co-expression of distinct shell proteins playing a kinetic race to truncate the growing cargo globule[62,63]. In the propanediol microcompartment specifically, the shell factors PduABK have all been strongly linked to cargo packaging with interactions supported along their interior-presenting convex surfaces[31,37,59,64–66]. Cargo valency and EP domain topology may be key considerations when packaging non-native cargoes to ensure robust network formation in addition to shell targeting and packaging. While any one single EP does not adhere to the canonical stickers and spacers architecture[61], higher order cargo oligomers (dimers, tetramers) act as an emergent topological mimic thereof. In addition, we cannot disregard that an in vivo environment would modulate the self-assembly properties due to a higher viscosity of the intracellular environment, which may explain the discrepancy between the in vitro and in vivo behaviors.

In addition to valency effects derived from protein quaternary structure, which exacerbate EP network formation (Supplemental Fig. 5E), the biophysical features of cargo subunits themselves may also influence condensation. For instance, we found that mScarlet-I3 cargo regularly displays a higher partitioning ability than mNeonGreen cargo in vivo (Supplemental Fig. 10), despite both being noted to be strictly monomeric with low aggregation tendencies[67]. However, increased localized concentration (leading to condensation) is a driving force for non-specific protein interactions, a mechanism which we proposed is exacerbated by the non-covalently crosslinking of EPs[68]. We do note that EPs are found on a variety of enzymes with different quaternary states, biophysical features, and biochemical roles[29]. Indeed, the history of synthetically implementing EP biotechnology in both shell-free and shell-bound systems argues in favor of their broad application for assembling diverse cargoes[36,69]. These findings are congruent with the natural role of EPs as a modular assembly peptide[5].

Our experimentation shows that salt bridge interactions, while not the predominant factor, help guide EP condensation. Knocking out the salt bridge makes cargo condensation both more sensitive to NaCl and lowers overall condensation propensity (Fig. 4C, D). These results suggest that EP-driven cargo nucleation events may be environmentally responsive to cellular conditions, including ions. This would not be without precedent, as biomolecular condensates have been found to be regulated by metabolites, ions, and environmental conditions[22,70,71]. Additionally, condensates themselves are recently being appreciated as also selecting for metabolites and small molecules[72]. Taken together, these notions suggest that prometabolosomes and EP-driven condensates could partition, and be influenced by, specific cofactors and metabolites during biogenesis[73–75]. Environmental responsiveness to redox status has been observed for CcmM-driven condensation[12] in β-carboxysome biogenesis but further investigations for metabolosomes are warranted.

## Comparison to other condensing peptides

There is great interest in developing small peptides that self-assemble into biomolecular condensates. Synthetic peptides have been developed with this in mind and have well-defined phase diagrams and controllable properties. However, many of these are intrinsically disordered, heavily utilize π–π stacking interactions, and/or rely on non-specific mechanisms for protein co-envelopment, which mechanistically differs from the EP domain[76–78]. Other synthetic and naturally implemented condensation systems rely on accessory partners for network formation, particularly nucleic acids[79–84]. These characteristics apply equally when also just considering naturally occurring systems in bacteria[1]. The closest analog to our knowledge is the PopZ microdomain, which has been adapted into the fully modular and

tunable PopTag[23,85]. Still, the EP domain is more compact than the PopTag and serves dual roles in cargo self-assembly and micro-compartment encapsulation. Futher, the occurrence of EPs N- and C-terminally fused to diverse enzymes suggests these are tractable in a wide range of chimeras[5,29]. These features make the EP domain standout and have high potential to serve as a platform for future biotechnological development.

## Applications and closing remarks

Metabolosomes can package an array of enzymatic functions within a wide array of microbes. Nature has supported this functional diversity by evolving a function agnostic, one-size-fits-all assembly approach. At its core lies the structural and functional features of the EP domain, which can be leveraged as a molecular tool to assemble disparate cargos in vitro and in vivo in a controlled, predictable manner. In vivo, these complexes may be used to scaffold target metabolic pathways for improved catalytic efficiencies (for example, substrate channeling). The applications for in vitro EP-driven phase separation are more diverse, for example: encapsulation of therapeutic molecules, stabilization of cell-free biomanufacturing systems, self-healing materials, among others. This work will help spur additional hypotheses on modes of assembly and rational design efforts using EP for biotechnology applications.

# Methods

## Molecular biology

A complete list of primers, synthetic gene fragments, and plasmid names used in this study can be found in the Supplemental Materials. Briefly, all cloning used the NEBuilder® HiFi DNA Assembly Cloning Kit (#E5520S), Q5® High-Fidelity 2× Master Mix (#M0492S) and/or KLD Enzyme Mix (#M0554S) offerings from New England Biolabs. Gibson assemblies were carried out using 0.02 pmol of backbone with a threefold molar excess of insert. All gene fragments were ordered from Twist Biosciences as codon optimized for *E. coli* and assembled into a pre-linearized pET11a backbone. All primers were purchased from IDT. Gibson assembly reactions were completed at 50 °C for 20 min and then immediately frozen at −20 °C until transformed into NEB5alpha chemically competent *E. coli* (#C2988J). Transformation was performed by incubating cells with assembly mix (<10% total volume) for 10 min on ice, heat-shocked at 42 °C for 30 s, then recovered in SOC media for 30 min at 37 °C prior to plating. Clone screening was performed on miniprepped plasmid DNA with whole-plasmid sequencing offered by Plasmidsaurus. PCR reactions were performed at 50 μL scale using a 15 s/kb extension time and appropriate primer annealing parameters. PCR reactions were treated directly with 1 μL of DpnI (#R0176S) to remove template DNA for 30 min at 37 °C followed by cleanup and concentration using the Zymo DNA Clean and Concentrator kit (#D4034) and quantified with a Qubit dsDNA BR kit (#Q32850) from Invitrogen. For PCR mutagenesis, 1 μL of successful PCR was directly mixed with KLD Enzyme Mix for 10 min at 25 °C and frozen until used directly for transformation.

## Protein expression and purification

Appropriate plasmids were transformed fresh into NEB T7 Express (#C2566H) competent *E. coli*. Single colonies were used to inoculate 4 mL of 2 × YT broth and carbenicillin antibiotic and grown overnight at 37 °C, 250 RPM. Overnight cultures were then used to inoculate 250–500 mL of 2 × YT supplemented with antibiotic in baffled flasks and grown to an optical density at 600 nm of -0.6–0.8 at 37 °C, 250 RPM. At that density, cells were briefly cooled to room temperature prior to the addition of IPTG to 100 μM and continued incubation at 20 °C and 250 RPM for 16–18 h. Cells were harvested the next day via centrifugation (5000 × *g*, 10 min) and frozen at −20 °C for at least 30 min. Thawed pellets were then resuspended at 5 mL/g wet cell pellet in NP40 High Salt Lysis Buffer (RPI, #N32000) supplemented

with 0.5 mM PMSF, 0.5 mg/mL egg white lysozyme, 5 mM MgCl$_2$ and 12.5 U/mL recombinant benzonase (Syd Labs, #BP4200). The suspensions were lysed at 25 °C for 30 min with constant agitation. Insoluble material was then removed by centrifugation (12,000 × *g*, 15 min, 4 °C). The supernatants were then decanted into new tubes and stored on ice while the pellets were resuspended in an additional 10 mL of lysis buffer (as above) and incubated for an additional 20 min. The insoluble matter was removed as before, and the supernatants were combined. The supernatants were then incubated at room temperature with 1 mL of PureCube 100 Ni-INDIGO agarose beads (#75105) equilibrated into Buffer A (20 mM Tris-HCl, 150 mM NaCl pH 8.0) on an end-over-end rocker for 30–60 min at room temperature. Proteins were then purified by gravity flow with 15 mL of washing (Buffer A + 25 mM imidazole) and 5–8 mL of elution (Buffer A + 500 mM imidazole). Proteins were then dialyzed into Buffer A such that remaining imidazole was <0.1 mM using 10 K MWCO Slide-A-Lyzer Dialysis Cassettes from Thermo Fisher (#66380). In some instances, protein was dialyzed into 20 mM Tris-HCl pH 8.0 without additional salt. Proteins were then concentrated and quantified by A$_{280}$ before normalization to 200 μM and flash freezing into liquid nitrogen to store for future use. SDS-PAGE profiles of purified proteins can be found in Supplemental Fig. 6.

## Turbidity assays

Turbidity assays were performed in 96-well plate format in quad-ruplicate and setup with the help of an Integra Assist Plus liquid handler in either standard (200 μL) or half-area (100 μL) plates. When assaying protein concentration effects on turbidity, stock sample was serially diluted into Assay Buffer (20 mM Tris-HCl, 150 mM NaCl pH 8.0) with appropriate mixing prior to adding an equal volume of Assay Buffer supplemented with 40% (w/v) PEG2k to reach a final value of 20%. When assaying the effects of crowding agent, an equal volume of PEG2k at a 2× final value in Assay Buffer was added to diluted protein solutions. Lastly, the effects of salt were assayed by first serially diluting a stock solution of 20 mM Tris-HCl, 5 M NaCl pH 8.0 with 20 mM Tris-HCl pH 8.0, before addition of appropriate protein (25 μM final concentration) and finally by addition of 20 mM Tris-HCl pH 8.0, 40% PEG2k. In all cases, turbidity was measured at 600 nm in a Biotek Synergy plate reader. In all cases, error bars represent the standard deviation from the reported mean.

## Fluorescence resonance energy transfer

Condensation reactions were prepared at room temperature in tripli-cate 100 μL volumes in a 96-well half-area plate. Briefly, reactions were performed in 20 mM Tris-HCl, 150 mM NaCl, 20% (w/v) PEG2k with 20 μM mNeonGreen (donor) and 5 μM mScarlet-I3 (acceptor). Reactions were mixed via pipetting and left to equilibrate for 5 min before reading turbidity and FRET in a Biotek Synergy plate reader. A control sample consisting of 20 μM mNeonGreen alone was used to subtract the non-specific FRET signal.

## Laser scanning confocal microscopy

For in vivo imaging, protein expression was carried out using the same conditions as when expressing protein for purification. Samples were harvested with centrifugation (3000 × *g*, 2 min) and resuspended in phosphate-buffered saline (PBS) pH 7.4. Samples were then spotted into a prepared 1.5% agarose pad in PBS, spread and dried for 5 min prior to imaging on an Olympus FV3000. For in vitro imaging of dro-plets, 5 μL of each sample was sandwiched between two coverslips and imaged directly. The 405 nm laser line was used to excited mTagBFP2 and 488 nm laser line for mNeonGreen, 561 nm laser line for mScarlet-I3 and. Images were colored in ImageJ and are presented without further modification beyond cropping. ImageJ was also used for quantification of partition ratios by measuring the mean gray value of background subtracted foci and non-foci regions within cells. In all cases, *n* = 100 for partition ratio calculations. Photobleaching

experiments, both in vivo and in vitro, were completed by imaging 5–10 s of pre-bleach sample at 0.5 s intervals prior to spot irradiation with the 488 nm laser for 0.5 s at 50% output. Images were then collected at 0.5 s intervals for 2 min. Bleaching quantification was performed in ImageJ.

## Computational simulations

For any given system, simulations were run in four independent replicates. PduP EP models were based on the observed solution structure[31] and modelled using Modeller[86], imposing a helical structure conformation. The helical structure was then equilibrated and simulated in a periodic box using the charmm36[87] force field and the Gromacs[88] molecular dynamics engine. Equilibrated peptides were later represented using the Martini coarse-grained force field in its version 2.2. This step was used to explore the dimerization process of two independent helices and to extract equilibrated configurations, which were backmapped into fully atomic structures using our developed pipeline[89]. The process was repeated to simulate trimers, tetramers, and pentamers. We also ran simulations of equilibrated trimeric bundles at varying peptide concentrations (0.02, 0.04, 0.08, and 0.1 mM, respectively) to characterize the formation of larger bundle aggregates. Simulations were run with the latest Martini 3 force field to prevent excessive aggregation, which can mislead conclusions[49]. Each system was composed of 80% water and 20% PEG. Finally, we built EP2 and EP3 tandem structures with Modeller to characterize self-EP association within the construct. Topologies were represented using an updated version of the Martini 3 to better match the dynamic properties of the disorder linker region[90].

All atomistic simulations followed the same protocols as provided here. Each backmapped peptide system (dimers to pentamers) was centered in a cubic box, hydrated, and neutralized with 150 mM KCL. The size of the box was chosen to create at least 1.5 nm of padding on each side along the largest atom-atom distance of the peptide. A constant temperature of 310 K was maintained using velocity Langevin dynamics[91] with a relaxation time of 1 ps. A constant isotropic pressure of 1 bar was maintained using the C-rescale approach with a relaxation time of 1 ps and compressibility of 4.5105 bar. Covalent bond lengths involving hydrogens were constrained using the SHAKE algorithm[92] with a tolerance of 10-6 nm. Water molecules were rigidified with SETTLE[93]. Lennard-Jones interactions were evaluated using a cut-off where forces smoothly decay to zero between 1.0 and 1.2 nm. Coulomb interactions were calculated using the particle-mesh Ewald) approach. In total, 4 simulations were performed for each system with an aggregated time of 4 μs.

Coarse-grained simulations with the Martini 2.2 version were run as previously described[94]. Simulations were carried out at 310 K using isotropic pressure coupling at 1 bar with a time step of 20 fs. Non-bonded Lennard-Jones potentials used a cutoff radius of 1.1 nm. The reaction field method was used for electrostatic calculations[95]. The cutoff distance for dielectric constant was set to 1.1 nm. Within the cutoff, the dielectric constant was 15, and beyond the cutoff it was infinite. The velocity Verlet algorithm[96] was used for integrating Newtonian equations. Temperature was controlled by a Langevin dynamics thermostat with a coupling time constant of 5 ps. Box pressure was controlled using a C-rescale method. For each CG MD production run, frames were saved every 100,000 time steps (every 2 ns). Configurations were categorized using the gmx cluster tool using the gromos approach. Simulations with Martini 3 were performed using the recommended parameters from the original work[49].

## Reporting summary

Further information on research design is available in the Nature Portfolio Reporting Summary linked to this article.

## Data availability

The data generated in this study is provided within the manuscript and in the Supplementary Information. All plasmids generated for this study are available upon request and after a Material Transfer Agreement with LANL. PDB codes 5LTR and 1GGX were used to generate graphics in Fig. 6. Source data are provided with this paper.

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

## Acknowledgements

The authors would like to thank Dr. Eric Young for his thoughtful commentary and feedback on this manuscript. This research used resources produced by the Los Alamos National Laboratory (LANL) Institutional Computing Program, which is supported by the U.S. Department of Energy National Nuclear Security Administration under Contract No. 89233218CNA000001 (C.A.L.). The authors (D.S.T., E.R., B.L.M., and C.R.G.E.) gratefully acknowledge the Laboratory Directed Research and Development (LDRD) program of LANL under project number 2024001DR.

## Author contributions

D.S.T.: conceptualization, investigation, methodology, visualization, writing–original draft, writing–review and editing. C.A.L.: investigation, methodology, visualization, writing–original draft, writing–review and editing. E.R.: investigation, methodology, writing–original draft, writing–review and editing. B.L.M.: conceptualization, funding acquisition, project administration, supervision, writing–original draft, writing–review and editing. C.R.G.E.: conceptualization, project management, supervision, writing–original draft, writing–review and editing.

## Competing interests

The authors declare no competing interests.
