## [Transparent Peer Review file · Nature Communications]

A blueprint for biomolecular condensation driven by bacterial microcompartment encapsulation peptides

Corresponding Author: Dr Cesar Gonzalez-Esquer

Version 0:

Reviewer comments:

Reviewer #1

(Remarks to the Author)

Summary:

The authors investigate how encapsulation peptides, which enable the sequestration of cargo into bacterial microcompartments, drive the formation of biomolecular condensates. To do so, they attach encapsulation peptides to several different types of fluorescent proteins. Using both in vitro and in vivo fluorescence microscopy, they examine how condensate formation and material properties depend on protein concentrations, salt concentration, and encapsulation peptide identity. They find that encapsulation peptides containing a specific sequence motif drive the formation of single- and multi-component condensates over a range of conditions. They also use molecular dynamics simulations to study the molecular origins of this robust assembly behavior. They find that, in addition to hydrophobic interactions, salt bridges stabilize the interface between encapsulation peptides (a previously unrecognized mechanism). Their simulations also suggest that the encapsulation peptides can self-assemble into bundles, which results in a nonlinear dependence of the critical concentration for phase separation on encapsulation peptide valency.

Assessment:

Publish after minor revisions. The authors have demonstrated that attaching encapsulation peptides to protein cargoes can serve as a novel and straightforward method for driving phase separation of several (non-bacterial microcompartment-associated) proteins, and thus may be a powerful general strategy for inducing phase separation of a wide range of proteins. They have thoroughly characterized the phase separation behavior over a wide range of conditions, several protein and encapsulation peptide types, and both in vivo and in vitro. They have also identified the molecular mechanism by which encapsulation peptides drive condensation. This work should be of interest to a diverse community, including biologists, chemists, and physicists studying biomolecular condensates and bacterial microcompartment assembly. I suggest the following minor revisions

Major points:

- The molecular dynamics simulations provide valuable insights into the molecular interactions that drive phase separation, but it is a little unclear how some of the simulations connect to the experiments. For example, the different bundle structures formed in simulations of four and five peptides are interesting, but there are no corresponding experiments. Could the authors comment on what the simulation results might imply for experiments that use four or five peptides? On the other hand, there are experimental results, but no simulation results, for two peptides. Would it not make sense to also perform simulations for two peptides and compare to the simulations with three peptides? It would be valuable to see a comparison of the number and strength of salt bridges and solvent-accessible surface area between the two and three peptide cases, and to compare these results to the experiments (could the simulations help explain the non-monotonic behavior of condensation with number of peptides?) I realize that the authors have already done quite a few simulations, but presumably the simulations of two peptides should be quite a bit faster than those for three, four, and five.
- It should be made clear in the main text what the resolution of the model being used is for the simulations; specifically that the dynamics were performed with the Martini force field.

- On a related note, could the authors provide additional details on how they assessed the convergence of the simulation distributions shown in Fig. 5C?

Minor point:

- There are several instances where the authors use the word “particles” to refer to condensates. This is confusing; the authors should stick with “condensates”.

Reviewer #2

(Remarks to the Author)

Reviewer #3

(Remarks to the Author)

The manuscript by Trettel et al describes that the encapsulation peptides (EPs) in driving cargo accumulation into biomolecular condensates. Through in vitro experiments and molecular dynamics simulations, the authors demonstrate that hydrophobic packing and electrostatic interactions stabilize trimeric EP bundles, enabling programmable liquid- or gel-like partitioning. However, this manuscript merely outlines a blueprint for the utilization of EPs, as the study does not provide any experimental data or evidence demonstrating the practical use of EPs in relevant biological or biotechnological contexts, which significantly weakens its impact. All experimental data in the manuscript are centered around the ability of EPs to drive the aggregation of fluorescent proteins into biomolecular condensates. Additionally, the methods employed in this study are relatively simple, and the data presented are somewhat limited in scope and depth. There are inconsistencies between the in vitro and in vivo experimental results, which need to be addressed to strengthen the validity of the findings.

Major Concerns:

1) Lack of practical applications:

The study would be significantly strengthened by including experiments that validate the functional utility of EPs beyond fluorescent protein aggregation. For example, demonstrating the encapsulation of enzymes, therapeutic molecules, or other biologically relevant cargo, as well as testing the performance of EP-driven condensates in improving catalytic efficiencies, would provide stronger evidence for the proposed biotechnological potential of EPs. Without such data, the proposed applications remain theoretical and lack convincing support.

2) Limited novelty:

Peptides for encapsulating biomolecules have already been extensively reported (for examples, PMIDs: 36702825, 40069227, 32747754, 39929988). Since the functions of EPs are analogous to these existing systems, it is necessary to compare them and experimentally validate the superiority of EPs. Specifically, what benefits do EPs offer over these well-established peptide-based systems? Addressing this would significantly enhance the novelty and impact of the study.

3) Validation of BMC localization for EP-driven cargo segregation:

In the in vivo experiments, the authors fused the EPs of PduP to an mNeonGreen tag and overexpressed this construct in bacteria, observing the formation of segregated mNeonGreen cargo foci. However, it remains unclear whether this design truly enables the segregation of protein cargo into bacterial microcompartments (BMCs). A previously published study (PMID: 32332738) investigated the subcellular distribution of PduP in *Salmonella enterica* serovar Typhimurium LT2, where PduP was fused with eGFP at its C-terminus and recombined into the chromosome. The localization patterns reported in that study appear significantly different from those observed in the current manuscript, particularly for the EP3 design. This discrepancy raises concerns about whether the observed foci represent true BMC-localized structures or non-specific aggregates. To strengthen their claims, the authors should provide additional evidence demonstrating that the fluorescent foci formed in their experiments are indeed localized within BMCs. Clarifying this point is critical to support the conclusion that EPs drive cargo segregation into BMCs.

4) Discrepancy between in vitro and in vivo results:

The manuscript highlights a notable discrepancy between the properties of biomolecular condensates formed in vitro and in vivo. Specifically, in vitro experiments show that EP-driven condensates exhibit gel-like properties, while in vivo experiments in bacterial cells reveal liquid-like states. Given that the intracellular environment is more complex than in vitro conditions, is it reasonable that EPs can drive programmable liquid-like partitioning in vivo? The discrepancy between in vitro and in vivo condensate properties should be thoroughly discussed to clarify the limitations and potential of the EP system.

5) Linking EP-driven condensation to bacterial aggresome formation

The manuscript demonstrates that EP-driven condensation in bacteria exhibits liquid-like properties. This finding aligns with a previous study (PMIDs: 34669478) revealing that bacterial aggresomes are formed through liquid-liquid phase separation (LLPS). It is surprising that the authors did not discuss or contextualize their findings within this broader framework. Incorporating a discussion of how EP-driven condensation relates to or differs from aggresome formation would significantly enhance the manuscript's scientific impact and provide a more comprehensive understanding of the observed phenomena.

Minor problems:

1) In the main text, the sentence "The droplets formed by EP-mNG, however, did not experience significant recovery over a qualitative 2-minute sampling (Figure 2E) indicating slow dynamic internal rearrangements indicative of a gel-like state" contains an error. The correct figure reference should be Figure 1E, not Figure 2E.

Reviewer #4

(Remarks to the Author)

This manuscript by Trettel et al. presents a rich and very comprehensive body of data from multiple perspectives on encapsulation peptides (EPs) from metabolosomes. The study substantially advances our understanding of how EP condensates function in metabolosome biogenesis and offers valuable insights for the rational design of EPs for future applications. In particular, I would like to highlight several key strengths of the manuscript:

1. While the majority of the study focuses on the EP from PduP, I appreciate that the authors broadened the scope (around Figure 4) to include EPs from multiple other species as well as the corresponding sequence from CcmN in the carboxysome biogenesis, which has been discussed quite extensively in recent literature. A bonus point was the experimental evidence of the different behavior between CcmN EPs and metabolosome EPs. Overall, including a broad set of EP sequences would alleviate the concern that the described phenomenon is only pertaining to PduP, and ensures that the study appeals to a broader audience.
2. The identification of a salt bridge by molecular dynamics (MD) simulations (Figure 3), subsequently supported by experimental data, is a particularly strong aspect of the study. This well-executed combination of computational and experimental approaches provides valuable mechanistic insight and deserves recognition.
3. An early concern when reading through the *in vitro* data was that the EP-driven condensates appeared gel-like rather than liquid-like. However, the authors addressed this elegantly by performing live-cell studies later in the manuscript (around Figure 6), demonstrating that these condensates exhibit liquid-like behavior *in vivo*.

Some of these points would have been major concerns of the manuscript, but the authors have already done experiments to address them. For all the above reasons, I strongly recommend this manuscript for publication in Nature Communications. Below, I offer suggestions that may help further strengthen the manuscript prior to publication:

- 1) One interesting observation (Figure 2) is that EPs fused to mScarlet-I3 alone do not form droplets. This suggests that EP-mediated phase separation may depend on the specific biophysical properties and surface characteristics of the cargo protein, which is unsurprising but highly relevant. In later *in vivo* studies (Figure 7 and Supplemental Figure 5), the authors also note differences between mScarlet-I3 and mNeonGreen, and in fact, made the comment that "the characteristics of the individual proteins themselves can influence partitioning." The author should expand on this discussion, and perhaps including simulations and/or leveraging known structural information on these fluorescent proteins to provide further mechanistic insights. Such information would be valuable for researchers seeking to design EP-cargo systems for future applications.
- 2) In Figure 4C, distinct structural morphologies are observed in the micrographs of PduE and EutC condensates. PduE tends to form fibril-like clusters, while EutC forms large, irregular aggregates. It would be helpful if the authors could speculate on the potential molecular or biophysical factors driving these differences.
- 3) For Figure 5D, the authors might consider extending the mutational analysis performed in Figure 3C (e.g., R11A) to include mutations of E16 (e.g., E16A or E16K). This would help directly test whether the proposed electrostatic interaction between R11 and E16 is critical for stabilizing the observed three-helix configuration.
- 4) In the section 'EP topology can alter partitioning and material state *in vivo*', the sentence "We quantified this effect by calculating a partition ratio..." introduces the partition ratio calculation somewhat late. Since cargo partitioning is already discussed in Supplemental Figure 2A and 2B in an earlier section, I suggest that the method of calculating partition ratios be introduced earlier in the manuscript, ideally when partitioning data first appears. This would help streamline the narrative and make subsequent references to the metric more intuitive for readers.
- 5) For Figure 6, the term 'emergent valency' could benefit from a clearer definition in the Discussion section. I encourage the authors to explicitly define this term and contrast it with 'intrinsic valency', which would help clarify its conceptual significance.
- 6) Including additional confocal micrographs would strengthen the manuscript. Specifically:
 - a) For Figure 1C, showing how EP-mNeonGreen condensate formation varies with protein concentration (e.g., threshold concentration for phase separation and progression from diffuse to condensed states) would be valuable.
 - b) For Figure 3C, providing representative micrographs showing the salt sensitivity of condensates under different ionic strengths would visually illustrate how electrostatic screening affects condensate formation, stability, and dissolution.
- 7) Please add scale bars to Figure 6E and Supplemental Figure 4B for completeness.
- 8) There is a small figure reference error: in the sentence "...a qualitative 2-minute sampling (Figure 2E) indicating slow dynamic internal rearrangements indicative of a gel-like state," the figure citation should refer to Figure 1E rather than Figure 2E.

Reviewer #5

(Remarks to the Author)

I co-reviewed this manuscript with one of the reviewers who provided the listed reports. This is part of the Nature Communications initiative to facilitate training in peer review and to provide appropriate recognition for Early Career

Researchers who co-review manuscripts.

Version 1:

Reviewer comments:

Reviewer #1

(Remarks to the Author)

The authors have addressed all of our comments and I believe it is ready for publication in Nat. Comm.

Reviewer #3

(Remarks to the Author)

The manuscript has been improved after the revision and it is now acceptable for publication.

Reviewer #4

(Remarks to the Author)

The authors have addressed all of my comments and have substantially improved the manuscript. Regarding comments 3 and 4, although the authors did not follow the suggestions exactly as proposed, their reasoning is sound. I have no further comments.

REVIEWER COMMENTS

Reviewer #1 (Remarks to the Author):

Major points:

The molecular dynamics simulations provide valuable insights into the molecular interactions that drive phase separation, but it is a little unclear how some of the simulations connect to the experiments. For example, the different bundle structures formed in simulations of four and five peptides are interesting, but there are no corresponding experiments.

- We value the reviewer's comment and have reworked part of the text incorporating new MD generated data. We expect these extra sets of simulations better complement the experimental evidence to cohesively and rationally providing conclusive remarks. Interplay between experiments and theory is shown in the following statements:
 - First, Multiscale (CG-AA) self-assembling simulations provide atomic detailed insights (residue pairing) of EP multimerization, data which was used to guide the experimental mutational analysis provided in Figure 3. Results provided in Figure 5 shed lights into the most stable helical bundle a key component that can lead condensation. Based on their stability, we conclude that a trimeric structure is the most stable associated conformer. We added new data coming from unbiased CG simulations using the latest Martini 3 forcefield to prevent excessive aggregation, thus preventing an unrealistic association. Agreeing with the limited AA sampling of EP bundles, CG simulations confirm that trimeric peptides are the most probable scaffold during the association.
 - Second, tandem-EPs (more than an EP helix) outperforms a single EP; however, 3 repeats marginally improve 2 EP's capacity, an indication that somehow a saturation has been reached. We ask whether tandem-EP self-quenching in the constructs may affect the cross-linking between independent mNEonGreen proteins, limiting the condensation. We characterized the dynamics of EP2 and EP3 constructs, quantifying the interaction between their connected Eps. This new simulated data indicates that connected EPS can dimerize/trimerize within the construct, reducing their propensity for inter-connectivity. In fact, EP3 constructs can interconvert allowing different association levels, with a maximum of three coordination sites. However, there are configurations that don't allowed further association (0 open binding sites). Although these simulations are not fully recapitulating the complexity of the tested constructs, it provides insights at molecular scale of the probable bundle association, which leads to hypothesize the larger and complex association mechanism as provided in Figure S3. We recognize that this representation is minimalistic, but it aids in explaining some of the experimental observations. One may imagine engaging EPs with other constructs creating an extendable network while still satisfying the maximum stoichiometry per construct. We agree that a

direct correlation would require measuring the association propensity of independent EPs (no linker), but it is beyond the scope of the manuscript.

- One note to explain why we didn't experimentally pursue higher copy numbers of tandem repeats is that we ran into increasing issues cloning with higher copy numbers. This is likely due to the repetitious nature of the sequences, which we did try to account for at the design stage. As such, we used MD simulations to attempt to access information about these higher-order designs which we could not make ourselves. These issues are now noted in the appropriate results text to better clarify our design choices and limitations thereof.

Could the authors comment on what the simulation results might imply for experiments that use four or five peptides?

- Concern has been addressed in the previous comment.

On the other hand, there are experimental results, but no simulation results, for two peptides.

- We disagree; in this case, designing the mutants was based on the simulation of dimers which highlighted the importance of hydrogen bonds. We have modified the text to better reach this conclusion in the manuscript.

Would it not make sense to also perform simulations for two peptides and compare to the simulations with three peptides? It would be valuable to see a comparison of the number and strength of salt bridges and solvent-accessible surface area between the two and three peptide cases, and to compare these results to the experiments (could the simulations help explain the non-monotonic behavior of condensation with number of peptides?) I realize that the authors have already done quite a few simulations, but presumably the simulations of two peptides should be quite a bit faster than those for three, four, and five.

- We have added an extra supporting image highlighting this concern. As mentioned in the manuscript, dimeric association exposes to much of hydrophobic surface as evidenced in Figure S6, and rendering this configuration unstable as evidenced by its rmsd. However, part of the association is stabilized by the hydrogen bonds discussed in the main manuscript. We plot the normalized number of hydrogen bonds and sasa per peptide, clearly suggesting that the trimer is most likely the preferable state in solution

It should be made clear in the main text what the resolution of the model being used is for the simulations; specifically that the dynamics were performed with the Martini force field.

- Text has been modified accordingly.

On a related note, could the authors provide additional details on how they assessed the convergence of the simulation distributions shown in Fig. 5C?

- Convergency was addressed by block averaging.

Minor point:

- There are several instances where the authors use the word “particles” to refer to condensates. This is confusing; the authors should stick with “condensates”.
 - We thank the reviewer for noting that we use several different phrasing to describe the same set of objects. On their suggestion, we have rectified this and stick to “condensates” where appropriate. The single instance where this is not changed is when describing the basis of turbidity assays, which are a general assay relying on scattering by any particle in solution, and is therefore not specific to condensates alone.
-

Reviewer #2 (Remarks to the Author):

Reviewer #3 (Remarks to the Author):

The manuscript by Trettel et al describes that the encapsulation peptides (EPs) in driving cargo accumulation into biomolecular condensates. Through in vitro experiments and molecular dynamics simulations, the authors demonstrate that hydrophobic packing and electrostatic interactions stabilize trimeric EP bundles, enabling programmable liquid- or gel-like partitioning. However, this manuscript merely outlines a blueprint for the utilization of EPs, as the study does not provide any experimental data or evidence demonstrating the practical use of EPs in relevant biological or biotechnological contexts, which significantly weakens its impact. All experimental data in the manuscript are centered around the ability of EPs to drive the aggregation of fluorescent proteins into biomolecular condensates. Additionally, the methods employed in this study are relatively simple, and the data presented are somewhat limited in scope and depth. There are inconsistencies between the in vitro and in vivo experimental results, which need to be addressed to strengthen the validity of the findings.

Major Concerns:

1) Lack of practical applications:

The study would be significantly strengthened by including experiments that validate the functional utility of EPs beyond fluorescent protein aggregation. For example, demonstrating the encapsulation of enzymes, therapeutic molecules, or other biologically relevant cargo, as well as testing the performance of EP-driven condensates in improving catalytic efficiencies, would provide stronger evidence for the proposed biotechnological potential of EPs. Without such data, the proposed applications remain theoretical and lack convincing support.

- In this manuscript, and as stated by the reviewer, we have characterized a system for generating protein condensates (“the blueprint”) which we now make available to the public. We are taken aback by the critic that we use “relatively simple” methods, as no research should need to push for complexity without need. It is our hope that the further additions (see Reviewer 1) will strengthen the manuscript and remove any perceived inconsistencies. Lastly, we have ongoing research that demonstrates the application of these peptides in i) biomanufacturing of petroleum replacements, and ii) in recovery of molecules from solution. We consider these to be of enough complexity and content that they warrant a separate manuscript, and we’d be happy to discuss with the editor if needed.

2) Limited novelty:

Peptides for encapsulating biomolecules have already been extensively reported (for examples, PMIDs: 36702825, 40069227, 32747754, 39929988). Since the functions of EPs are analogous to these existing systems, it is necessary to compare them and experimentally validate the superiority of EPs. Specifically, what benefits do EPs offer over these well-established peptide-based systems? Addressing this would significantly enhance the novelty and impact of the study.

- The reviewer states that our system of study lacks intrinsic novelty because other systems exist and/or have been designed to support biomolecular condensation. In fact, we believe the articles the reviewer points out reflects the novelty and importance of the EP and demonstrate that they are not analogous, despite the reviewers claims. For instance, among the articles the reviewer points out:
 - 36702825, 40069227, and 3277754 all rely heavily on aromatic residues while EPs do not. Further, these all are intrinsically disordered protein-based systems. Meanwhile, EPs are helical, and we have supporting evidence (for future publications) that the EP linker is largely irrelevant. This showcases the mechanistic differences which nevertheless lead to similar results.
 - Further, the peptides in 36702825 and 40069227 were never shown fused to a cargo protein and were never demonstrated in vivo.
 - 39929988 does showcase a synthetic condensation system, but it relies on an RNA counterpart. This is mechanistically entirely different from EPs, which support self-assembly. EPs, therefore, are far more modular for controlling protein assembly.
 - EPs from bacterial microcompartments are also unique among other described prokaryotic phase separation systems (PMID 33186556). The closest analog for self-assembly is the PopTag (PMID 36163138), which is 4x the length of an EP domain and requires IDR elements which EPs do not. Further, EPs are more widespread among bacteria than PopZ domains. Further, the EP domain is dual-purpose, unlike the PopTag, as it results in self-condensation and assembly in microcompartments depending on the context.
 - The purpose of our study was to demonstrate the principles required for the EP to undergo condensation – a goal which we achieved and stands on its

own merits. This is a similar goal to the other papers brought up by the reviewer, none of which did any comparison to other systems.

- These points highlight the novelty of the EP among phase separation inducing peptides. As such, a new Discussion section has been added which contextualizes EPs within the larger peptide-based condensation literature which we believe will enrich readers.

3) Validation of BMC localization for EP-driven cargo segregation:

In the *in vivo* experiments, the authors fused the EPs of PduP to an mNeonGreen tag and overexpressed this construct in bacteria, observing the formation of segregated mNeonGreen cargo foci. However, it remains unclear whether this design truly enables the segregation of protein cargo into bacterial microcompartments (BMCs). A previously published study (PMID: 32332738) investigated the subcellular distribution of PduP in *Salmonella enterica* serovar Typhimurium LT2, where PduP was fused with eGFP at its C-terminus and recombined into the chromosome. The localization patterns reported in that study appear significantly different from those observed in the current manuscript, particularly for the EP3 design. This discrepancy raises concerns about whether the observed foci represent true BMC-localized structures or non-specific aggregates. To strengthen their claims, the authors should provide additional evidence demonstrating that the fluorescent foci formed in their experiments are indeed localized within BMCs. Clarifying this point is critical to support the conclusion that EPs drive cargo segregation into BMCs.

- We believe the reviewer is misinterpreting the current work and the work of Yang et al. 2022. The localization patterns are different because they are entirely different systems. Yang et al. were studying colocalization patterns of the entire PduP enzyme to the PDU bacterial microcompartment in its native host, which known to produce dispersed microcompartment foci. In contrast, we were expressing the EP of PduP fused to a fluorescent reporter in *E. coli* without co-expression of a microcompartment. Further, common lab strains of *E. coli* do not have microcompartments of their own to interfere with interpretations. As such, the different systems expectedly produce different colocalization patterns.
- We have added text to the last paragraph of the introduction to clarify that our experiments are outside of a microcompartment context.

4) Discrepancy between *in vitro* and *in vivo* results:

The manuscript highlights a notable discrepancy between the properties of biomolecular condensates formed *in vitro* and *in vivo*. Specifically, *in vitro* experiments show that EP-driven condensates exhibit gel-like properties, while *in vivo* experiments in bacterial cells reveal liquid-like states. Given that the intracellular environment is more complex than *in vitro* conditions, is it reasonable that EPs can drive programmable liquid-like partitioning *in vivo*? The discrepancy between *in vitro* and *in vivo* condensate properties should be thoroughly discussed to clarify the limitations and potential of the EP system.

- Text has been amended in the Discussion to state that we believe that the higher viscosity environment inside of cells may modulate self-assembly and therefore contribute to the different behaviors *in vitro* and *in vivo*.

5) Linking EP-driven condensation to bacterial aggresome formation

The manuscript demonstrates that EP-driven condensation in bacteria exhibits liquid-like properties. This finding aligns with a previous study (PMIDs: 34669478) revealing that bacterial aggresomes are formed through liquid-liquid phase separation (LLPS). It is surprising that the authors did not discuss or contextualize their findings within this broader framework. Incorporating a discussion of how EP-driven condensation relates to or differs from aggresome formation would significantly enhance the manuscript's scientific impact and provide a more comprehensive understanding of the observed phenomena.

- There are ~dozen LLPS systems defined in bacteria (PMID 33186556), of which the aggresome is just one. As such, in response also to the reviewers note #2, we have added a discussion section contextualizing EP condensation within the wider protein condensation literature. We focus our comparison to the PopTag system which we believe is the closest analog.

Minor problems:

1) In the main text, the sentence "The droplets formed by EP-mNG, however, did not experience significant recovery over a qualitative 2-minute sampling (Figure 2E) indicating slow dynamic internal rearrangements indicative of a gel-like state" contains an error. The correct figure reference should be Figure 1E, not Figure 2E.

- We thank the reviewer for catching this, and it has now been corrected.

Reviewer #4 (Remarks to the Author):

This manuscript by Trettel et al. presents a rich and very comprehensive body of data from multiple perspectives on encapsulation peptides (EPs) from metabolosomes. The study substantially advances our understanding of how EP condensates function in metabolosome biogenesis and offers valuable insights for the rational design of EPs for future applications. In particular, I would like to highlight several key strengths of the manuscript:

1. While the majority of the study focuses on the EP from PduP, I appreciate that the authors broadened the scope (around Figure 4) to include EPs from multiple other species as well as the corresponding sequence from CcmN in the carboxysome biogenesis, which has been discussed quite extensively in recent literature. A bonus point was the experimental evidence of the different behavior between CcmN EPs and metabolosome EPs. Overall, including a broad set of EP sequences would alleviate the concern that the described phenomenon is only pertaining to PduP, and ensures that the study appeals to a broader audience.

2. The identification of a salt bridge by molecular dynamics (MD) simulations (Figure 3), subsequently supported by experimental data, is a particularly strong aspect of the study. This well-executed combination of computational and experimental approaches provides valuable mechanistic insight and deserves recognition.

3. An early concern when reading through the in vitro data was that the EP-driven condensates appeared gel-like rather than liquid-like. However, the authors addressed

this elegantly by performing live-cell studies later in the manuscript (around Figure 6), demonstrating that these condensates exhibit liquid-like behavior in vivo.

Some of these points would have been major concerns of the manuscript, but the authors have already done experiments to address them. For all the above reasons, I strongly recommend this manuscript for publication in Nature Communications. Below, I offer suggestions that may help further strengthen the manuscript prior to publication:

1) One interesting observation (Figure 2) is that EPs fused to mScarlet-I3 alone do not form droplets. This suggests that EP-mediated phase separation may depend on the specific biophysical properties and surface characteristics of the cargo protein, which is unsurprising but highly relevant. In later in vivo studies (Figure 7 and Supplemental Figure 5), the authors also note differences between mScarlet-I3 and mNeonGreen, and in fact, made the comment that “the characteristics of the individual proteins themselves can influence partitioning.” The author should expand on this discussion, and perhaps including simulations and/or leveraging known structural information on these fluorescent proteins to provide further mechanistic insights. Such information would be valuable for researchers seeking to design EP-cargo systems for future applications.

- The reviewer recommends increased discussion on the effects of cargo on EP-driven condensation. We agree that this would be beneficial. As such, we now add a passage in the Discussion noting the effects of cargo on condensation. We note that mNeonGreen and mScarlet-I3 are both known to be monomeric and highly soluble in vivo, and thus it is difficult to pinpoint exact reasons for their differences. Still, we argue that the history of EP synthetic implementation to this point, including shell-free (PMID: 26969252) and shell-bound systems (PMID: 40187678), argues in favor of their broad applicability.
- Along these lines, we have evidence of EPs triggering biomolecular condensation of several non-model proteins which are diverse in function and are being pursued for separate publications.

2) In Figure 4C, distinct structural morphologies are observed in the micrographs of PduE and EutC condensates. PduE tends to form fibril-like clusters, while EutC forms large, irregular aggregates. It would be helpful if the authors could speculate on the potential molecular or biophysical factors driving these differences.

- We thank the reviewer for their suggestion to postulate on the possible biophysical differences that lead to different EP-driven structures. We now speculate, in the discussion section, on the tentative importance of residue positions (like E16 in PduP) not explicitly tested by us that may help to explain the weaker condensation, and likely aggregation, of PduE and EutC EPs.

3) For Figure 5D, the authors might consider extending the mutational analysis performed in Figure 3C (e.g., R11A) to include mutations of E16 (e.g., E16A or E16K). This would help directly test whether the proposed electrostatic interaction between R11 and E16 is critical for stabilizing the observed three-helix configuration.

- We thank the reviewer for this suggestion and agree it would generally be meaningful to directly probe the E16 amino acid. However, we are choosing to abstain from that currently for the following reasons:
 - To expedite the publication process, since it would take several weeks to design, create, and test these mutations.
 - During review, we did anticipate and design a new mutant swapping the charges in the PduP EP (E7R/R11E), now included in Figure 3. We believed designing this mutant would proactively help further cement the antiparallel nature of the salt bridge originally anticipated in Figure 3B. Indeed, the sequence of this design looks like that of the PduD EP and maintains the wild-type PduP condensation propensity (Figure 3D), albeit with a heightened salt sensitivity (Figure 3C). This tentatively agrees with the notion of R11-E16 interactions since a charge swap to R11E would create charge conflict at that position. While new analysis of the trimer bundles (Supplemental Figure 5) continues to show E16 as an accessory factor, our other data strongly suggests that the primary contributors are (1) the hydrophobic core and (2) the antiparallel salt bridge.
 - The weak, but likely real, contribution of the E16 position we believe is also reflected in the sequence alignment presented in Figure 4A. This position is variable among the EPs we screened and seem to tentatively correlate with their condensation propensity. Acidic residues (PduP, PduD) correlate with strong condensation strength, polar residues (EutC) are in the middle, while basic residues (PduE) are the weakest. This is related to our comments on point (2) above.
 - These matters are now discussed in detail in the Discussion section. We attempt to be clear when our existing data supports, but does not directly prove, the importance of the E16 position which can be an area for future research.

4) In the section ‘EP topology can alter partitioning and material state in vivo’, the sentence “We quantified this effect by calculating a partition ratio...” introduces the partition ratio calculation somewhat late. Since cargo partitioning is already discussed in Supplemental Figure 2A and 2B in an earlier section, I suggest that the method of calculating partition ratios be introduced earlier in the manuscript, ideally when partitioning data first appears. This would help streamline the narrative and make subsequent references to the metric more intuitive for readers.

- The reviewer notes that our partition ratio calculation should narratively be described earlier in the text than current due to our reference to it in an earlier supplemental figure. While we understand this notion, we believe that since it’s in reference to a supplemental figure, it’s best introduced first alongside its first introduction in the main text. As a compromise, we now describe the partition ratio calculation within the caption for that specific supplemental figure.

5) For Figure 6, the term ‘emergent valency’ could benefit from a clearer definition in the Discussion section. I encourage the authors to explicitly define this term and contrast it with ‘intrinsic valency’, which would help clarify its conceptual significance.

- The reviewer notes that our usage of the terms “intrinsic” and “emergent” valencies could be better defined for readability. Indeed, these terms have clear definitions within the condensation literature. Intrinsic valency arises from a single molecule having >1 binding site, while emergent valency arises from multiple subunits assembling to endow the final assembly with >1 binding site. These definitions have been added to the appropriate discussion section with references for further reading.

6) Including additional confocal micrographs would strengthen the manuscript. Specifically:

a) For Figure 1C, showing how EP-mNeonGreen condensate formation varies with protein concentration (e.g., threshold concentration for phase separation and progression from diffuse to condensed states) would be valuable.

- We believe that this is represented later in the text, specifically Figure 5B as the EP1 design is same construct as in Figure 1, just renamed to match the nomenclature used for EP2 and EP3. We believed that it narratively fits better when paired with comparisons to the EP2 and EP3 designs.
- Further, we did not image samples at low enough concentrations to establish a threshold or saturation concentration. Experimentally, it is difficult as the droplets get increasingly small and sparse with lower concentrations and ~3 μM was the lowest we screened with confidence. Still, we now add text related to Figure 5B explicitly noting that a threshold concentration could not be established but may lay in the low micro- to high nano-molar range.

b) For Figure 3C, providing representative micrographs showing the salt sensitivity of condensates under different ionic strengths would visually illustrate how electrostatic screening affects condensate formation, stability, and dissolution.

- We thank the reviewer for this suggestion and agree the visual illustration would be useful. We completed this using the PduP EP:mNG under the same salt concentrations used in Figure 3C. The imaging performed shows, with increasing NaCl, a reorganization of condensates into aggregates which then dissolve with enough NaCl. These results have been noted in the main text and images can be found in Supplemental Figure 2.

7) Please add scale bars to Figure 6E and Supplemental Figure 4B for completeness.

- These changes have been added. In addition, missing scale bars in Supplemental Figure 5B have also now been added.

8) There is a small figure reference error: in the sentence “...a qualitative 2-minute sampling (Figure 2E) indicating slow dynamic internal rearrangements indicative of a gel-like state,” the figure citation should refer to Figure 1E rather than Figure 2E.

- This misattribution has been corrected.

Reviewer #5 (Remarks to the Author):

Other Changes:

Demonstration of heterotypic co-condensation of cargoes with different EPs

- While engaging with peers in the field on this work while under review, some had mentioned that they would like to see explicitly if different EPs can be used to co-condense multiple cargoes. Currently, we only demonstrate condensation of multiple cargoes using the PduP EP. In response to these conversations, we captured microscopy evidence of mNeonGreen (fused to the PduD EP) and mScarlet (fused to the PduP EP) co-assembling into condensates *in vitro*. These two EPs were chosen as proof-of-concept as they are the most robust natural sequences we screened. **This data is now presented in Figure 2** and the text has been updated appropriately. This data confirms heterotypic condensation derived from multiple different EPs which aligns with their known role in co-assembling multiple cargoes within microcompartments. We believe including this additional data will greatly enrich readers without warranting much extra scrutiny.

Additions to Supplemental Tables

- Several designs were accidentally left out of Supplemental Tables 1 and 3, which has now been rectified.